# Teaching deep networks to see shape: Lessons from a simplified visual world

**Christian Jarvers** *, **Heiko Neumann**

Institute for Neural Information Processing, Ulm University, Ulm, Germany

* christian.jarvers@uni-ulm.de

## Abstract

Deep neural networks have been remarkably successful as models of the primate visual system. One crucial problem is that they fail to account for the strong shape-dependence of primate vision. Whereas humans base their judgements of category membership to a large extent on shape, deep networks rely much more strongly on other features such as color and texture. While this problem has been widely documented, the underlying reasons remain unclear. We design simple, artificial image datasets in which shape, color, and texture features can be used to predict the image class. By training networks from scratch to classify images with single features and feature combinations, we show that some network architectures are unable to learn to use shape features, whereas others are able to use shape in principle but are biased towards the other features. We show that the bias can be explained by the interactions between the weight updates for many images in mini-batch gradient descent. This suggests that different learning algorithms with sparser, more local weight changes are required to make networks more sensitive to shape and improve their capability to describe human vision.

**Data Availability Statement:** All data and code used for data generation, model training and evaluation is available on a GitHub repository at https://github.com/cJarvers/shapelearningtheory.

## Author summary

When humans recognize objects, the cue they rely on most is shape. In contrast, deep neural networks mostly use local features like color and texture to classify images. We investigated how this difference arises, using images of simple shapes like rectangles and the letters L and T, combined with color and texture features. By testing different feature combinations, we show that some networks are generally unable to learn about shape, whereas others could learn to recognize shapes in isolation, but ignored shape if another feature was present. We show that this bias for color and texture arises from the way in which networks are trained: by averaging the learning signal over many images, the training algorithm favors simple features that are relatively similar in many images and removes sparser, more varied shape features. These insights can help build networks that are more sensitive to shape and work better as models of human vision.

We have also used Zenodo to assign a DOI to the repository: doi.org/10.5281/zenodo.10850391.

**Funding:** The author(s) received no specific funding for this work.

**Competing interests:** All authors declare that they have no competing interests to disclose.

## Introduction

Rapid advances in deep learning have created new opportunities to understand brain function [1–5]. Deep neural networks can now be trained to perform tasks like image classification or language generation at a higher level than any other model class and, in sufficiently narrow settings, they can even outperform humans [6, 7]. For neuroscientists who aim to explain how the brain gives rise to behavior, having models that can perform behaviors at this level of complexity is a crucial advantage [1]. In addition, deep neural networks were once inspired by and take their name from the biological neural networks studied by neuroscientists and therefore retain some similarity to the brain: they are made up from simple computational units that are connected in complex ways. Therefore, deep networks can express both mid-level hypotheses about what functions individual units compute and how they have to be connected, and high-level hypotheses about how experience (in the form of training data and learning rules) shapes processing [5]. This approach has been remarkably successful, especially in modeling visual processing, where deep networks have been used to predict cortical representations with high accuracy [8–10] and even to generate artificial stimuli to control responses in target neurons [11].

A key criticism of using deep networks as models in neuroscience is that their performance may be deceptive [12]. Even though a deep network may assign similar labels to images as a human participant in an experiment, it may use entirely different mechanisms and strategies to do so. While deep networks share some similarities with visual cortex, the latter makes use of many structural design principles that have not yet been adopted in deep networks, such as feedback connections or lateral grouping [13]. Indeed, several lines of evidence indicate that deep networks process images very differently from humans: they are susceptible to adversarial attacks [14, 15], less robust to image corruptions [6, 16], and do not show evidence of perceptual grouping phenomena that play key roles in human perception [12]. One key difference between human vision and deep networks is that humans strongly rely on shape when judging which category an object belongs to, whereas neural networks preferentially rely on texture [17, 18] and similar surface-level properties [19]. In fact, networks seem to be largely unable to process the global shape of objects in a human-like manner [19–21].

The recently formulated "neuroconnectionist" research strategy [5] views these differences not as a fundamental problem, but as an opportunity to investigate directions of further improvement. If we can understand why shape processing differs so drastically between deep networks and human vision, we may gain insight into the components and mechanisms that are required for human-like shape understanding.

Evidence for a lack of shape processing in deep networks comes from several lines of research (see [21] for a review). While deep networks are able to use some shape information, e.g., to classify silhouettes, they are relatively unaffected by image distortions that disrupt global shape, such as scrambling of silhouette parts [17, 22]. When shape and another property, such as color [19] or even the intensity of a single pixel [23], are equally predictive of image class, deep networks typically learn to use the other feature. Similarly, networks trained on natural images exhibit a "texture bias" [18]. When these networks are tested on cue conflict images, i.e., stylized images which have the shape of one class (e.g., the outline of a cat) with the texture of a different class (e.g., resembling elephant skin), networks choose the class of the texture in the vast majority of cases [18].

This phenomenon of texture bias on cue conflict images has received considerable attention in the machine learning literature and several methods have been proposed to mitigate it. These include explicit training on stylized images [18], stronger image augmentations [24], and adding custom network components [25]. However, increasing shape bias on cue conflict

images may not automatically lead to human-like shape processing—it is only one out of several relevant metrics [21]. As a case in point, networks trained on stylized images to have a high shape bias seem to remain unable to process global object shape [20, 22].

Although the shape bias of neural networks has been documented extensively, there is not yet a clear explanation for why it arises. However, two candidate hypotheses can be distilled from the literature:

**Hypothesis 1:** Shortcut learning. Since networks are trained to minimize a loss function (e.g., to classify images), they find the simplest possible feature to solve the task [16]. If other features do not provide additional predictive merit, they are ignored.

**Hypothesis 2:** Deficiencies in architecture. In this view, current networks are missing some component that enables them to process shape effectively. For example, mechanisms for grouping or amodal completion may be required [12, 20].

However, each of these hypotheses leaves something to be explained. In the case of shortcut learning, the original question—why networks rely on texture rather than shape—is replaced by another question: why is recognizing textures easier to learn for deep networks than recognizing shape? Conversely, if there is a deficiency in current network architectures, what is it? Which component is missing exactly and how would it enable shape processing?

These questions are difficult to answer, because *shape* and *texture* are ill-defined concepts, at least in natural images. We may know intuitively what the shape of a cat is and we may be able to recognize it in an image, but since we lack a formal definition of the "cat-shapedness" of an image, we cannot test whether a network computes "cat-shapedness". Several techniques have been used to manipulate shape and texture in natural images, such as stylization [18], use of silhouettes [17], and shape scrambling [17, 22]. This can be used to probe selectivity for shape and texture experimentally. However, none of these techniques is perfect, since each alteration in shape also has an effect on textures and vice versa [21].

One way to solve this issue is to use artificial images, in which shapes and textures are generated parametrically and can therefore be controlled exactly. For example, [19] generated images in which colorful patches formed simple shapes. They compared how well humans and neural networks could learn to distinguish categories that depended on different features such as shape, color of object segments, or color of a single patch. While humans relied strongly on shape, networks preferentially classified images by other features, such as color or average size.

Here, we follow a similar approach but focus on the question why deep networks fail to learn shape-based classification. We design artificial image datasets in which simple shape, color, or texture features determine which class an object belongs to. We use these datasets in a series of four experiments to test the two hypotheses. First, by training and testing the networks on different feature combinations, we show that a variety of networks exhibit the same biases towards color and texture as on natural images. Second, we show that some of these networks fail to learn the classification at all when they are trained from scratch on a dataset where shape is the only available feature, indicating that there is an architectural deficiency (**hypothesis 2**). However, even networks that are able to learn shape-based classification in isolation fail to do so when another feature (color or texture) is available, consistent with shortcut learning (**hypothesis 1**).

Third, to understand how this bias against shape arises during learning, we analyze the dynamics and structure of the neural tangent kernel (NTK) [26] of networks during training. We compare the NTK of a network during training to reference kernels of networks with known feature selectivity and show that the direction of gradient updates is more aligned with color and texture features than with shape. To understand how this alignment arises, we

analyze the structure of the NTK. Specifically, we identify groups of images which are similar to each other according to the similarity function expressed by the neural tangent kernel. This means that the network outputs for these images are related: changing how the network responds to one image will also affect how it responds to the others. Finally, we examine whether images in these groups tend to belong to the same class. We find that this is the case for NTKs of color- and texture-sensitive networks, but not of shape-sensitive networks. We hypothesize that such groups dominate the dynamics of gradient descent learning, explaining why networks are biased to color and texture.

Fourth, we show that reducing the alignment between color- and texture-selective gradients enables networks to learn shape-based classification. This suggests that the bias for local features, such as color and shape, arises from the dynamics of gradient descent learning. To train shape-selective deep networks, a different learning algorithms may be necessary.

## Results

### Experiment 1: Networks show color/texture bias on artificial images

We constructed four datasets of artificial images, in which each image showed a single, simple object on a gray background. Similar to the approach in [19], each dataset comprised two image classes and class membership was predicted by two features: the shape of the object and the color or texture of the object (see Fig 1). For example, the first dataset consisted of colorful rectangles. One class contained images of horizontal rectangles (i.e., rectangles which were wider than they were high) of red or blue color. The other class contained images of vertical rectangles (i.e., rectangles which were higher than they were wide) of green or magenta color. Thus, either feature was sufficient to distinguish the classes—a network could learn to classify rectangle orientation irrespective of color, it could learn to classify the color of a rectangle and ignore its orientation, or it could learn a mixture of both features. We call this dataset `Color Rectangles`.

The other datasets followed a similar scheme. In the second dataset (`Striped Rectangles`), the shapes were the same as in the first dataset (horizontal and vertical rectangles), but the second feature was texture, not color. Each rectangle contained a sine grating texture and the classes were distinguished by the orientation of this texture: class 1 had forward-slanted stripes, whereas class 2 had backward-slanted stripes. In the third dataset (`Color L-or-T`), each image contained two bars which could either form an L-shape (class 1) or a T-shape (class 2). The L-shaped objects had a red or blue color, whereas the T-shaped objects had a green or magenta color. The fourth dataset (`Striped L-or-T`) used the same L- and T-shapes as dataset 3 and the same texture classes as dataset 2 (slanted sine gratings). Example images from these dataset are shown in Fig 1. Details about the stimulus generation are described in the methods section.

Since these datasets have simple, precisely defined feature dimensions, we can use them to test how biases for certain features arise. However, this is only useful if networks trained on these datasets show the same biases as on natural images, i.e., a preference for color or texture over shape. Thus, our first goal was to test if networks trained on these datasets show this predicted color- and texture-bias.

To this purpose, we trained several networks on each dataset. We then evaluated the performance of each network on several test sets (see Methods): a shape-only version of the dataset in which the color or texture feature was absent (e.g., grey rectangles instead of colorful ones), a color- or texture-only version in which the shape feature was absent (e.g., colorful squares), and a conflict version in which the assignment of colors or textures to classes was inverted.

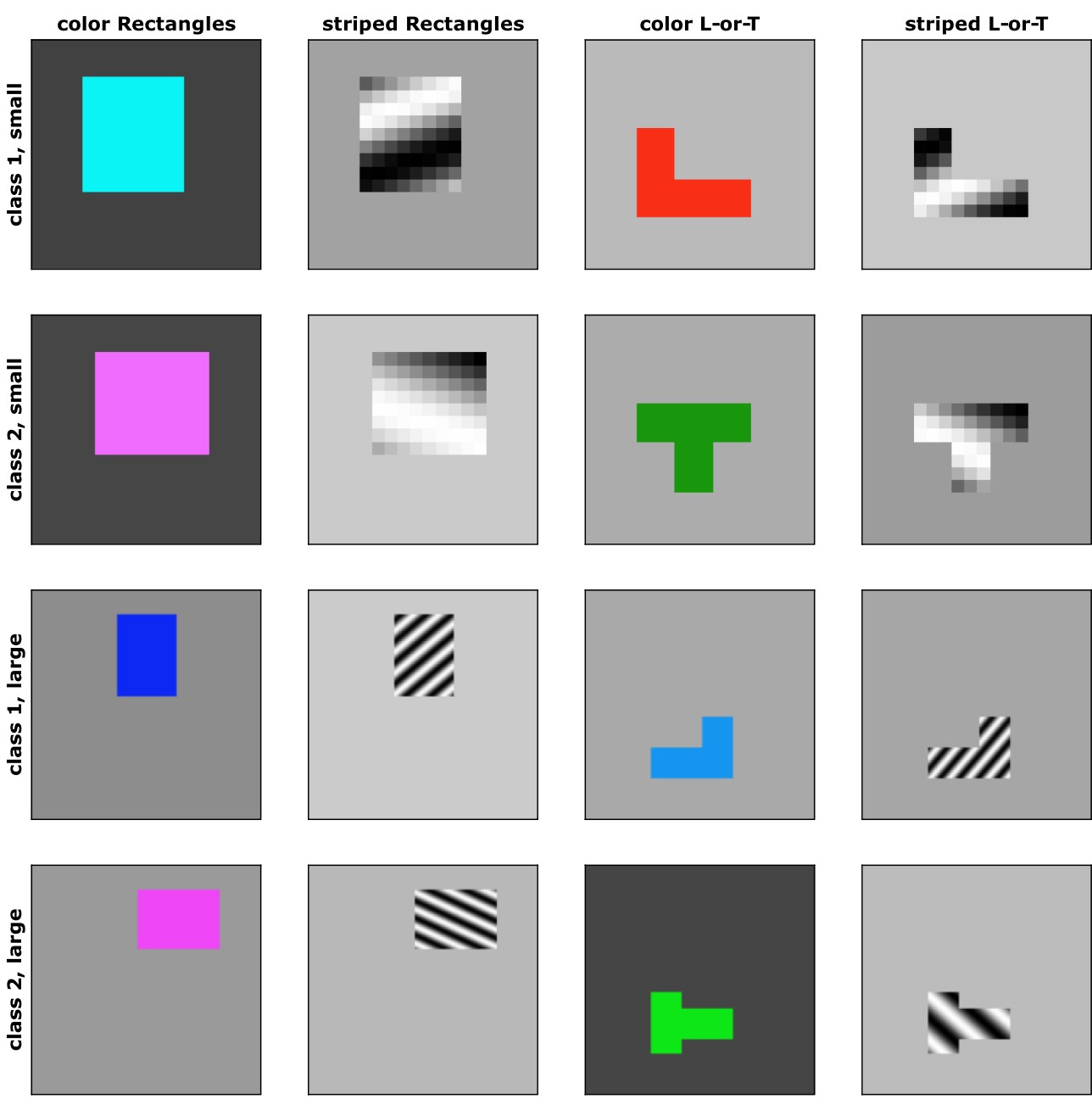

**Fig 1. Example images from datasets.** Example images from the four artificial image datasets (one dataset per column). Each row shows examples from one of the two classes. The top two rows show images from the small ($18 \times 18$ pixel) versions, whereas the two bottom rows show images from the large ($112 \times 112$ pixels) versions.

We tested a range of different networks to see if results were consistent across architectures. Fig 2 shows an overview of which networks were trained on which datasets. We chose networks that are widely used in the literature and represent different architecture types, including *VGG-19* [27] as an example of a convolutional network, *ResNet-50* [28] as an example of a residual network, *ViT-B-16* [29] as an example of a vision transformer, and *Swin-T* [30] as an example of a hybrid transformer architecture. In addition to these **standard networks**, we

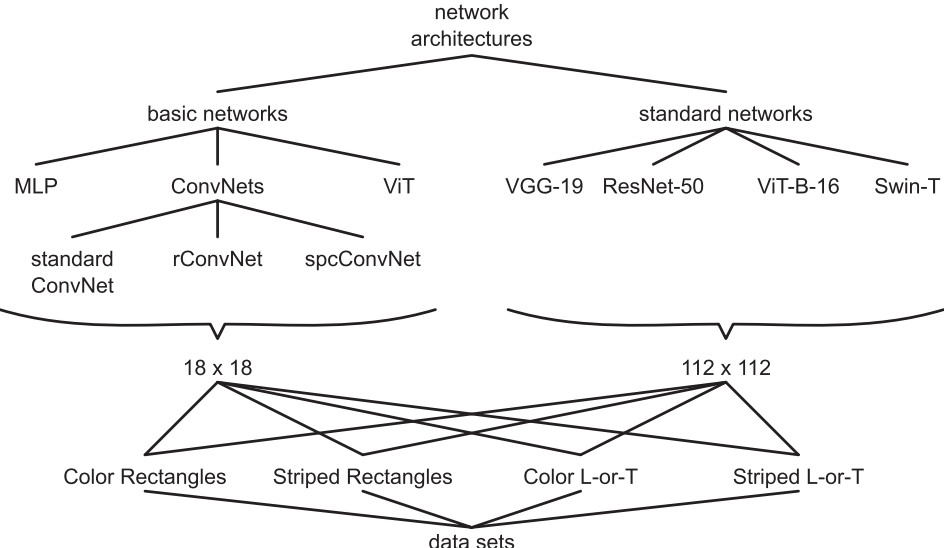

**Fig 2. Overview of networks and datasets.** We evaluated several basic network architectures on small (18 × 18 pixels) versions of our four image datasets. We also evaluated several standard network architectures on larger (112 × 122 pixels) versions of the same four image datasets.

designed a set of smaller networks. While we train the standard networks on images with 112 × 112 pixels, we train these smaller, **basic networks** on 18 × 18 pixel versions of our data-sets. This allows us to conduct more compute-intensive analysis (Experiment 3 and 4) more efficiently. The basic networks comprised a multi-layer perceptron (*MLP*), a convolutional network (*ConvNet*) and a vision transformer (*ViT*), as well as two convolutional networks with modifications inspired by biology: one with recurrent connections (*rConvNet*) and one with spatial competition (*spcConvNet*). For a detailed description of each architecture, see Methods.

Since the task is relatively simple, we expected each network to reach about 100% accuracy on the training set. If a network *learns to classify by shape*, we would expect high accuracy on the shape-only images, chance-level accuracy on the color-only images, and high accuracy on the conflict images (since shape-based classifications are counted as correct). In contrast, if a network *learns to classify by color or texture*, we would expect high accuracy on the color-/tex-ture-only images, chance-level accuracy on shape-only images, and low accuracy on conflict images. To test if the accuracy of a network was significantly above or below chance level for a given dataset, we ran the training 10 times with different random initial weights and compared the resulting distribution of accuracy values to 0.5 (chance level) with a sign test.

As shown in Fig 3, all standard networks exhibited the performance of a color-/texture-selective classifier that ignores shape: their accuracy was at ceiling on the training data and on the the color/texture-only test sets. They performed at chance level (50%) on the shape-only test set and classified nearly all images by color/texture on the conflict test set, resulting in near 0% accuracy. The only exception to this pattern was the *ViT-B-16* on `Striped Rectangles`, which performed above chance level on the shape-only test set ($M = 3$, $p = 0.031$). However, since the average accuracy was only 51%, it is unlikely that this result reflects real sensitivity to shape.

We performed the same experiment with the basic networks on the small version of our image datasets. The results are shown in Fig 4. The overall pattern of results was similar, with most networks (*MLP*, *ConvNet*, *rConvNet*, *ViT*) performing near ceiling on the training set

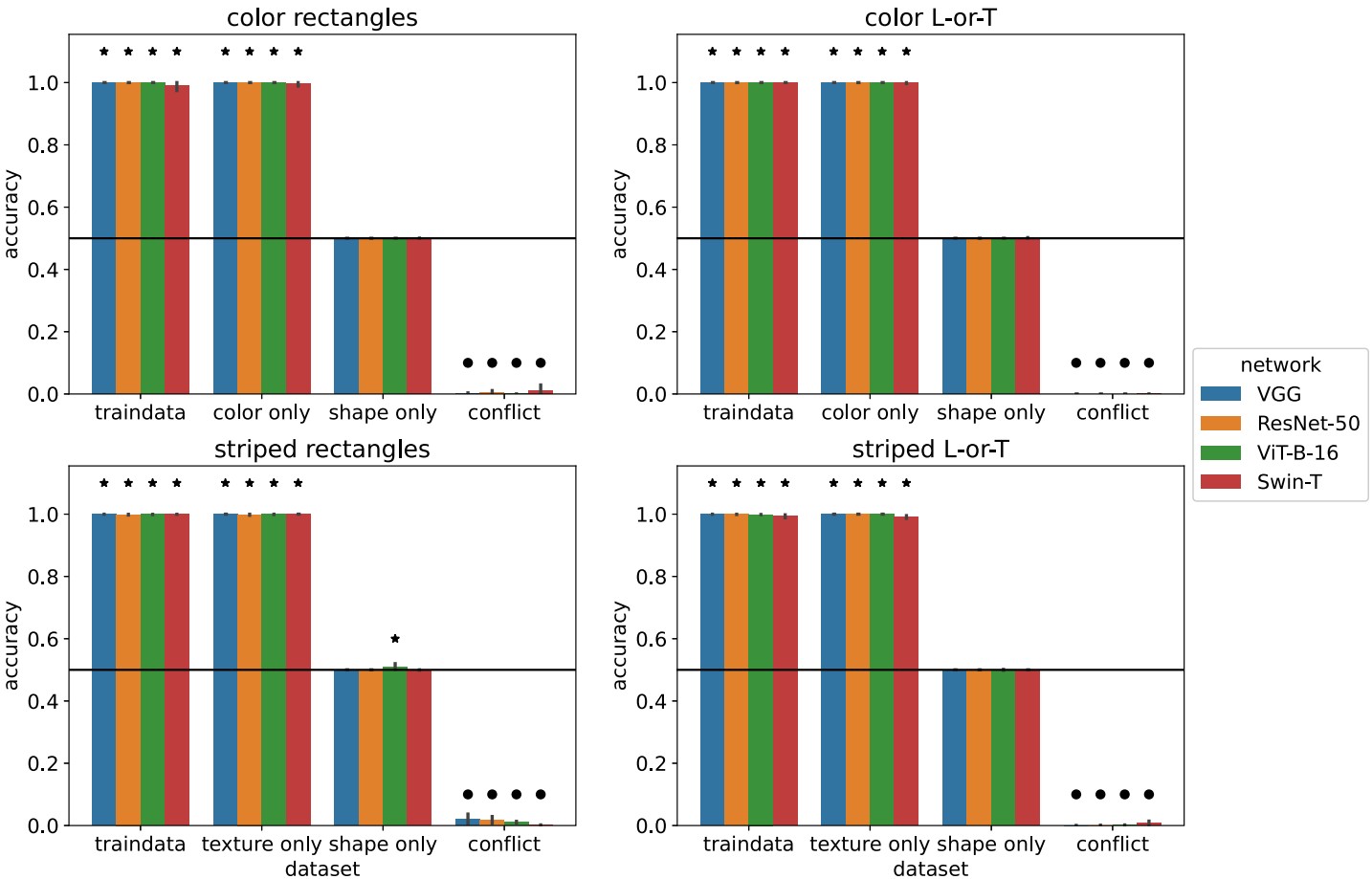

**Fig 3. Performance of standard networks.** Accuracy of standard networks after training on 112 × 112 pixel images. Images in the training set had both features (shape and color / texture). Error bars indicate 95% confidence intervals estimated from 10 training runs with different random initial weights. Stars indicate that accuracy is significantly above chance level (horizontal line) according to a sign test. Circles indicate that accuracy is significantly below chance level.

and on color-/texture-only test data, at or close to chance level on the shape-only test set, and near 0% accuracy on the conflict test set. In contrast to the standard networks, there were several instances of above-chance performance on shape-only test sets. In most of these cases, accuracy was only slightly above the chance level of 50% suggesting that the networks only developed a very limited shape selectivity. For example, networks trained on the texture datasets (`Striped Rectangles` and `Striped L-or-T`) probably developed detectors for oriented contrast. On the shape-only datasets, such detectors respond most strongly at the outline of the shape and this may suffice to bias the networks to respond correctly in some cases, e.g., because a horizontal rectangle has more horizontal than vertical contrast.

The only network that showed evidence of shape selectivity was *spcConvNet*. This network differs from the other convolutional networks by incorporating spatial competition: nearby locations in each feature map compete for activation (see Methods). This suppresses activations in homogeneous areas, enhances activations at unique feature locations, and makes the network representations sparser. *spcConvNet* was the only network with an accuracy above 60% on shape-only test sets with 86.5% on `Color Rectangles`, 82.5% on `Striped Rectangles`, and 63.9% on `Color L-or-T`. On the shape-only test set of `Striped L-or-T` it performed at 59.2% accuracy. *spcConvNet* was also the only network that was more

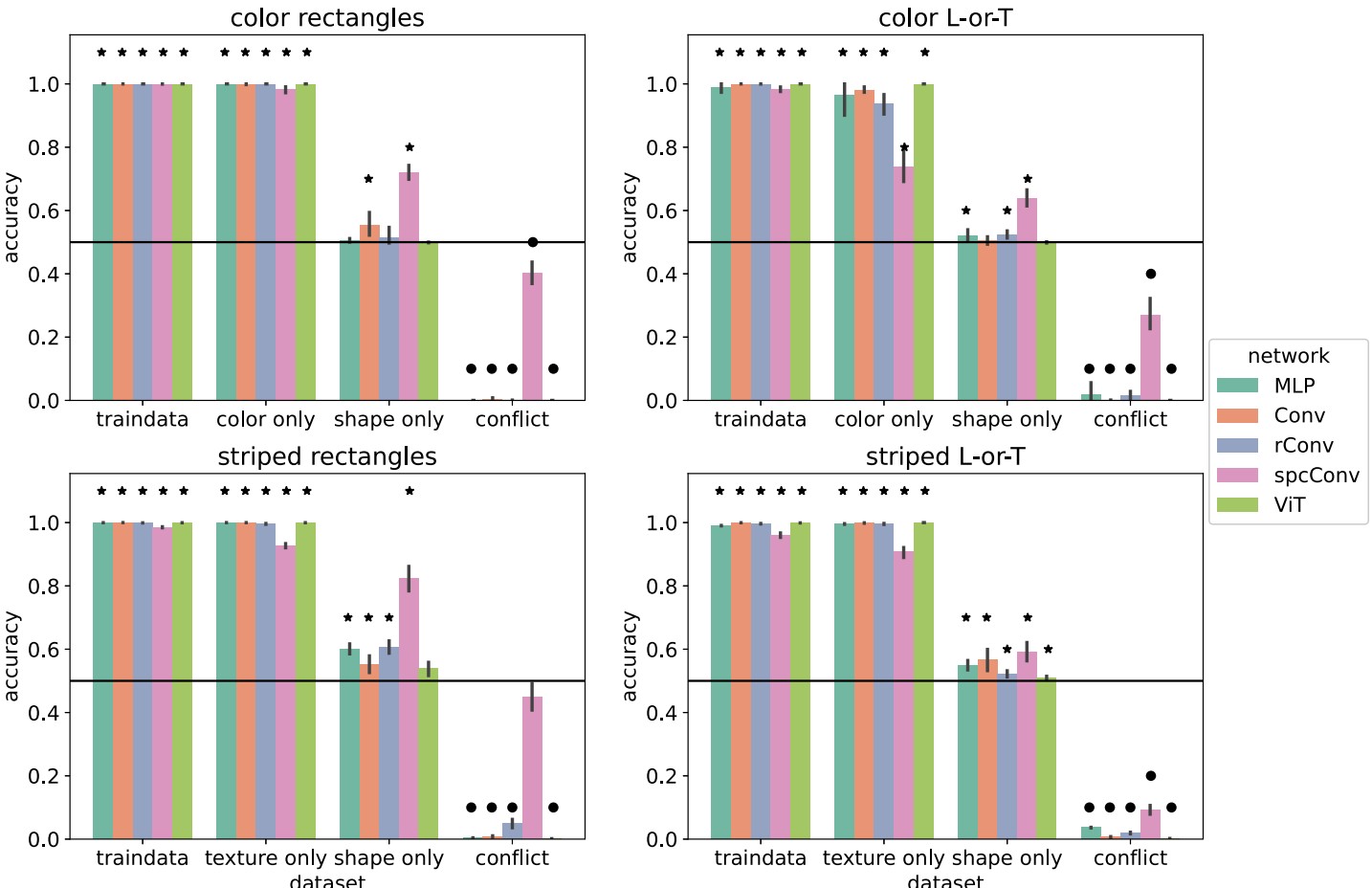

**Fig 4. Performance of basic networks.** Accuracy of basic networks after training on $18 \times 18$ pixel images. Images in the training set had both features (shape and color / texture). Error bars indicate 95% confidence intervals estimated from 10 training runs with different random initial weights. Stars indicate that accuracy is significantly above chance level (horizontal line) according to a sign test. Circles indicate that accuracy is significantly below chance level.

likely to classify conflict images according to shape, not color on the conflict test set of `Color Rectangles` (accuracy 68.6%, $M = 5$, $p = 0.002$). Furthermore, it showed no significant preference on the conflict test set of `Striped Rectangles` (accuracy 34.4%, $M = -2$, $p = 0.344$). Its performance was also above chance on all color- and texture-only test sets. This indicates that *spcConvNet* learned a mix of shape and color/texture features.

In summary, the majority of networks showed a clear bias towards color or texture on our artificial images. Thus, our image datasets can serve as a minimal visual domain to investigate how these biases arise.

## Experiment 2: Networks can learn to classify by shape

Having established that networks show a color/texture bias on our simple image datasets, our first research question was as follows: *Are the networks able to classify the image datasets by shape?* If networks are able to use the shape features in our datasets, their bias towards color and texture in experiment 1 is evidence for shortcut learning, supporting hypothesis 1. On the other hand, if networks are unable to learn to use the shape features in our datasets, this indicates a defect in network architecture, supporting hypothesis 2. Notably, an architectural defect

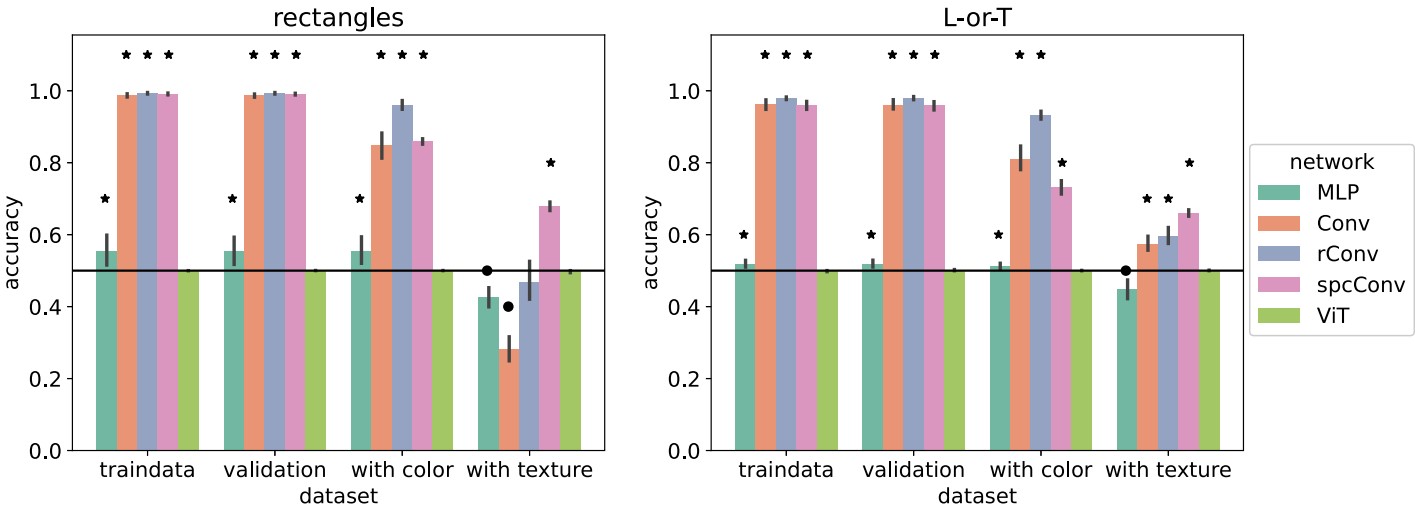

**Fig 5. Performance of basic networks trained on shape-only images.** Accuracy of basic networks after training on 18 × 18-pixel shape-only image datasets. Error bars indicate 95% confidence intervals estimated from 10 training runs with different random initial weights. Stars indicate that accuracy is significantly above chance level (horizontal line) according to a sign test. Circles indicate that accuracy is significantly below chance level.

does not have to mean that networks cannot learn to use any shape features, only that there are some shape features that networks are unable to learn. There is compelling evidence that deep networks can use some shape features, e.g., their ability to classify silhouettes [22]. Nevertheless, there may be some shape features that they are generally unable to learn.

We tested this question by training all networks on the shape-only version of each dataset, i.e., to classify uniform gray rectangles as horizontal or vertical and to classify uniform gray L and T shapes. We report accuracy on the training data, on a test dataset with the same shape properties but generated with a different random seed (e.g., with different random background grayscale levels), as well as on the colorful/textured versions of the datasets. The results for basic networks are shown in Fig 5 and for standard networks in Fig 6.

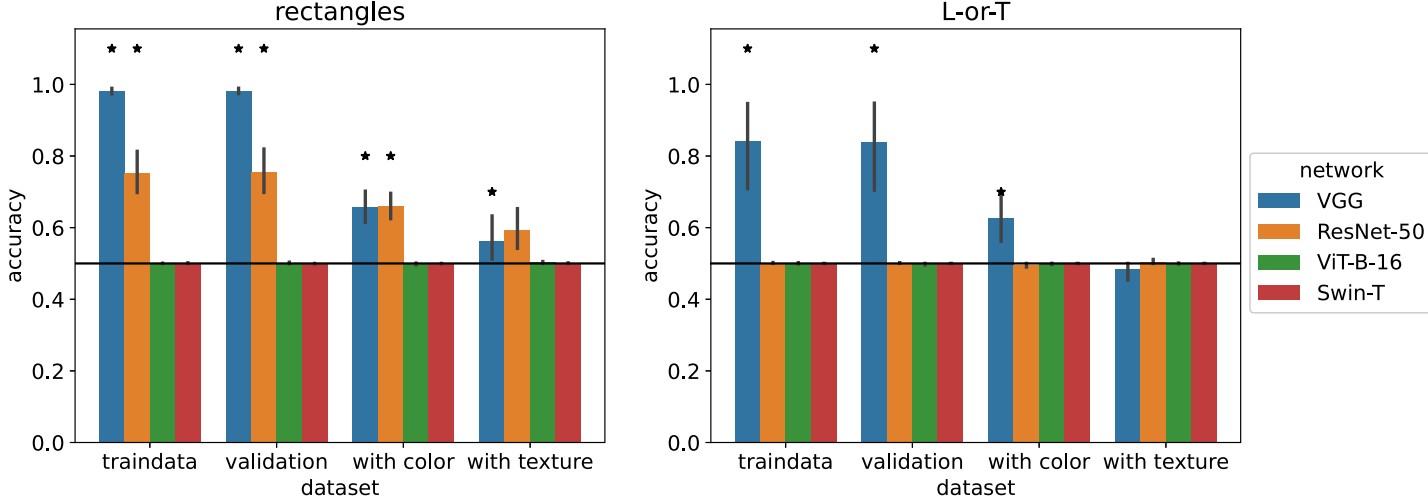

**Fig 6. Performance of standard networks trained on shape-only images.** Accuracy of standard networks after training on 112 × 112-pixel shape-only image datasets. Error bars indicate 95% confidence intervals estimated from 10 training runs with different random initial weights. Stars indicate that accuracy is significantly above chance level (horizontal line) according to a sign test. Circles indicate that accuracy is significantly below chance level.

The ability of networks to learn the shape-only classification task differed markedly depending on the architecture. Convolutional networks (*ConvNet*, *rConvNet*, and *spcConvNet*) performed near 100% accuracy on the training and test data, demonstrating that they could learn the task. When tested on images with color, they performed slightly worse but still closer to ceiling than to chance performance. This demonstrates that the features they learned are robust even outside of the original training distribution, which only contained grayscale images. On images with textures, performance was markedly lower, even below chance for *ConvNet* on `Striped Rectangles`. This is likely because the networks were only trained on texture-free images and learned convolution weights that responded to contrast (to detect object boundaries), which then also responded to sine grating textures. Nevertheless, the performance of *ConvNet*, *rConvNet*, and *spcConvNet* demonstrates that these networks could, in principle, learn to classify the datasets using shape alone.

The *MLP* showed a similar pattern of results, but at a lower performance level of about 55.4% on shape-only rectangles. On `shape-only L-or-T` it performed at 51.9%, hardly above chance level. This shows that the *MLP* can use some shape information, but it may also explain the preference for color or texture: if the *MLP* can classify all images correctly by using color, but only about 55% by using shape, relying on color is the better strategy.

The vision transformer performed at chance level (50%) on both datasets, with no variation between runs. The same was true for *ViT-B-16* and *Swin-T* on larger images. This suggests that there may be an underlying problem with the vision transformer architecture that reduces its ability to learn shape features. However, when we repeated the experiment with transformers pre-trained on ImageNet, both *ViT-B-16* and *Swin-T* were able to learn both shape-only classification tasks (see Fig E in S1 Appendix).

Similar to the small convolutional networks, *VGG-19* and *ResNet-50* were able to classify shape-only rectangles with an accuracy of 98.2% and 75.5%, respectively. However, on the `shape-only L-or-T` dataset, *VGG-19* only achieved an accuracy of 84% and *ResNet-50* failed to learn.

In summary, the ability of networks to learn shape-based classification did show a strong dependence on architecture. The *MLP* showed some capacity for shape-based classification, but to a much lower degree than when color or texture features were available. The vision transformers failed to learn when only shape information was available. In contrast, convolutional networks were able to learn the shape-only variants of each dataset. Thus, these networks are capable of learning shape-based classification. Nevertheless, with the exception of *spcConvNet*, all convolutional networks show a bias for color or texture over shape. We focus our further analyses on the basic *ConvNet* architecture to understand how this bias arises.

### Experiment 3: Bias in ConvNets arises from learning dynamics of gradient descent

So far, we have shown that convolutional networks are able to classify our image datasets by shape, but fail to learn shape-based classification if another feature (color or texture) is sufficient to predict the correct class. In order to understand how this bias arises, we examined the learning dynamics of our networks through the lens of the neural tangent kernel (NTK).

The neural tangent kernel $\Theta_f(x_i, x_j; \theta)$ characterizes the learning dynamics of a neural network $f(x; \theta)$ [26], where $x$ is an input image and $\theta$ are the trainable network parameters. Specifically, it can be thought of as a matrix with one coefficient for each pair of images $x_i$ and $x_j$ that describes how similar the gradients for these two images are. In this sense, the kernel describes a feature space in which some images are close to each other, while others are distant from each other. At the same time, the NTK describes a direction in the space of possible functions

that the network can instantiate (with different parameters). During training with gradient descent, the network parameters $\theta$ are adjusted such that the network function $f(x; \theta)$ changes and moves through this function space. Crucially, the direction of this movement is exactly the direction described by the neural tangent kernel [26]. This property is especially useful in the limit of infinitely wide neural networks, because in this limit the NTK remains constant during training. Therefore, training can be understood as a simple movement along a straight line through function space, where the direction of that line is determined by the neural tangent kernel [26]. For networks with finite width, the NTK changes during training as the network learns features from the data [31–33]. This means that training does not simply follow a straight line through function space. Nevertheless, at each point during training the NTK still describes the direction in which the network evolves and therefore describes the *direction* of what the network is learning [26]. Furthermore, the *structure* of the kernel (i.e., which images are similar in its feature space) reflects the features that the network has learned [31, 33].

In order to characterize when and how the bias for color and texture arises during training we compared the NTK of a network during training to two reference NTKs: one from a shape-selective network and the other from a color- or texture-selective network. We obtained these reference NTKs by training a network with the same architecture on shape-only and color-/texture-only images, respectively. For example, if the network under investigation was a *ConvNet* trained on `Color Rectangles`, the shape-selective reference NTK was generated by training a *ConvNet* on grayscale rectangles, and the color-selective NTK was trained on colorful squares. We trained each reference network for 30 epochs before calculating the NTK on a test set of `Color Rectangles`. We then trained 10 versions of the *ConvNet* with different random initializations on the full dataset (with color and shape features) and calculated the NTK on the same test set after each step of gradient descent. We calculated the similarity of each of these training NTKs with both reference NTKs (see Methods), resulting in two trajectories of NTK similarity. These trajectories are shown in Fig 7.

For all datasets, the NTK of the *ConvNet* trained on the full dataset was more similar to the color- or texture-trained network than for the shape-trained network. Thus, each step of gradient descent pushes the network more towards a color- or texture-based solution than towards shape selectivity.

Why is the NTK of the network trained on the full dataset more similar to the NTKs of the color- or texture-trained networks than to the NTK of the shape-trained network? To answer this question we examined the structure of the shape-, color-, and texture-trained NTKs. The NTK contains a coefficient for each pair of images in the training dataset and this coefficient expresses how similar the gradient vectors for these two images are. The structure of the NTK as a whole encodes pairwise similarities between all images. Thus, to understand its structure we can ask which images are similar according to the kernel. We identify clusters of such similar images by using the kernel as an affinity matrix for spectral clustering (see Methods). In effect, we cluster the images according to their similarity in the similarity space described by the NTK. We vary the number of clusters from 2 to 50. For each resulting set of clusters, we compute the adjusted Rand index [34] to test how well these clusters align with the image classes, i.e., whether images within one cluster typically belong to the same class. The results are shown in Fig 8.

For the *color NTK*, there was a high agreement between NTK clusters and image classes, with a peak at four clusters, both on the `Color Rectangles` and `Color L-or-T` datasets. In both cases, clusters in the *shape NTK* had a low agreement with the image classes, irrespective of cluster number. To test if this difference was significant, we took the maximum value of the adjusted Rand index for each of 10 independent training runs and compared the resulting distribution of scores for color-trained NTKs with the distribution for shape-trained

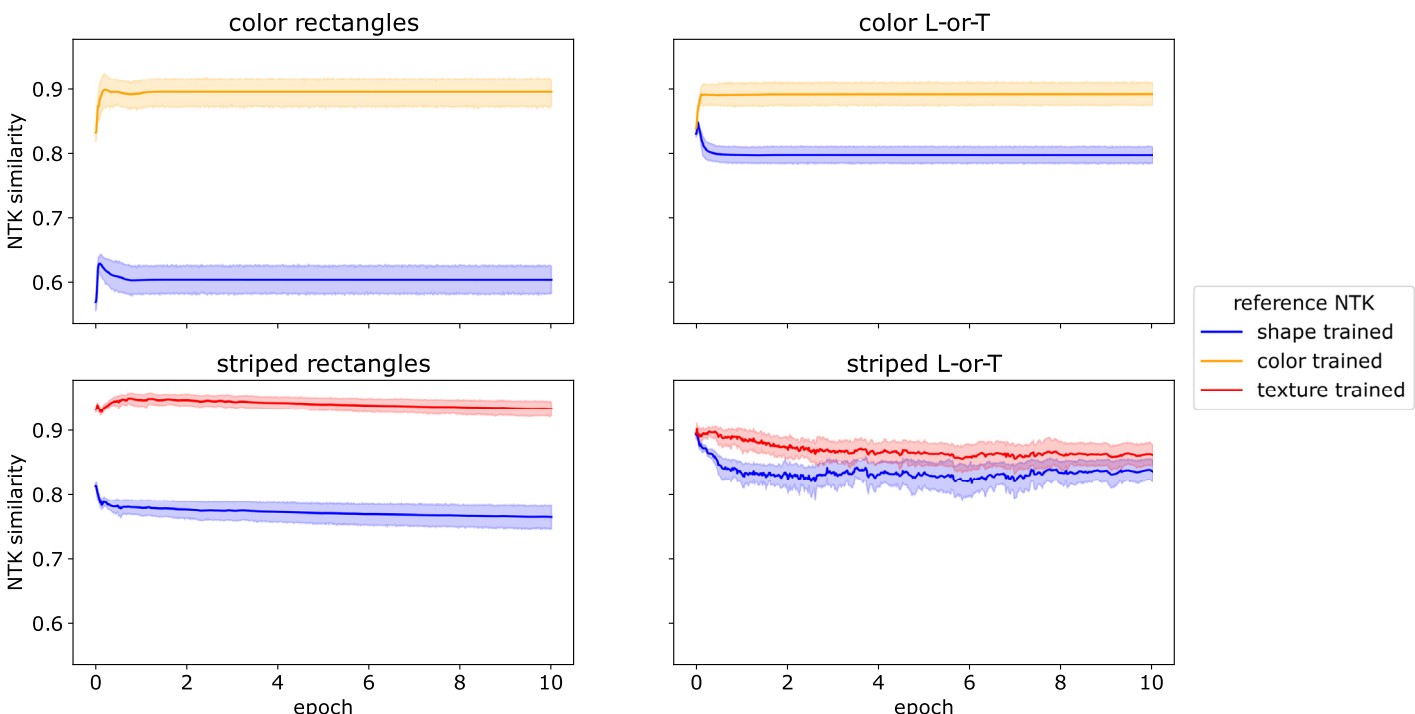

**Fig 7. NTK similarity for *ConvNet*.** Similarity between neural tangent kernels (NTK) of trained *ConvNet*s and of shape-selective and color-/texture selective *ConvNet*s throughout training.

NTKs with a Brunner-Munzel test [35, 36]. The test indicated a significant difference between the distributions both on `Color Rectangles` ($\hat{p}^*(18) = \infty$, $p < 0.001$) and on `Color L-or-T` ($\hat{p}^*(18) = \infty$, $p < 0.001$). The test statistic is $\infty$ due to the fact that all values for the color NTK were higher than those for the shape NTK.

For the *texture NTK*, the match between image clusters and class labels was lower and peaked later than for color. Nevertheless, the agreement was significantly higher for the texture NTK than for the shape NTK, both on `Striped Rectangles` ($\hat{p}^*(18) = \infty$, $p < 0.001$) and on `Striped L-or-T` ($\hat{p}^*(18) = \infty$, $p < 0.001$). As on the color datasets, clusters based on the *shape NTK* had a low agreement with image classes, independent of the number of clusters.

These results show that there are groups of images which belong to the same class and for which a color-selective or texture-selective network receives very similar weight updates. For a shape-selective network, such groups of images are the least common. This may explain why networks are biased towards color and texture: since training uses mini-batch gradient descent, the weight updates are averaged over many images. The weight update for a single image may increase selectivity for several features. However, if increasing the selectivity for one feature on multiple images requires similar changes to a subset of the network weights, the weight updates for these images will add up during the averaging operation. In contrast, if increasing the selectivity for another feature requires changing one subset of weights on one image and another subset of weights on another image, these weight updates will be averaged out. Our results suggest that color and texture belong to the former kind of feature, whereas shape belongs to the latter.

As a simple example of how these learning dynamics work, we can consider how an extremely simple convolutional network could learn the `Color Rectangles` task. Let us

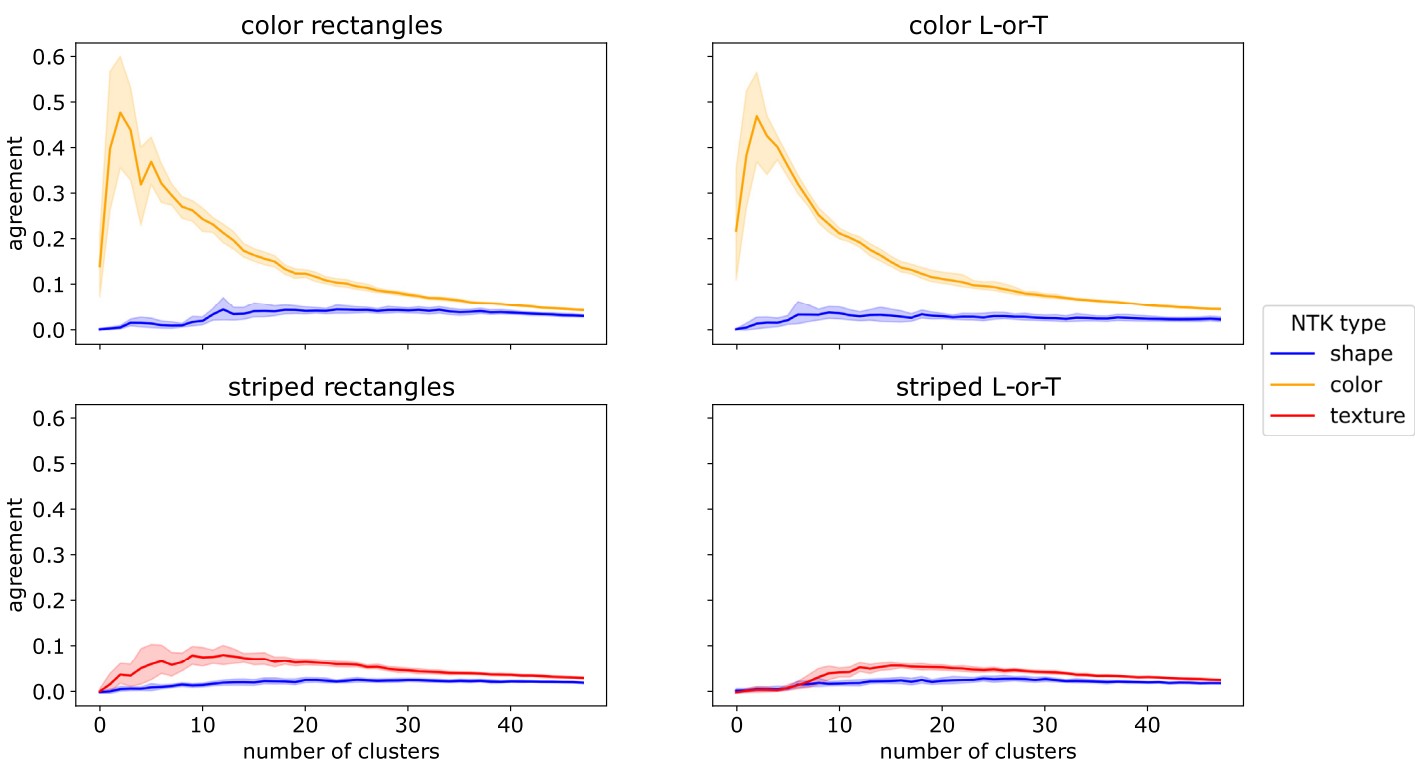

**Fig 8. NTK cluster analysis for *ConvNet*.** Agreement (adjusted Rand index) between ground truth labels and clusters of images with high similarity in neural tangent kernels (NTKs) trained to be selective to a single feature. Shaded areas show 95% confidence intervals over 10 independent training runs.

assume that the network only has one convolutional layer, followed by a ReLU non-linearity, global average pooling, and a fully connected layer with two output units. This network can solve the task in (at least) two different ways. Firstly, the convolutional layer could learn to detect colors and the fully connected layer could connect each color detector to the correct class. Secondly, the convolutional layer could learn to detect horizontal and vertical contrast and the fully connected layer could calculate the difference between the overall amount of horizontal and vertical contrast, since a horizontal rectangle should contain more horizontal contrast than a vertical one and vice versa. Assuming that one filter is randomly initialized with a weak selectivity for red color and another is randomly initialized with a weak selectivity for vertical light-to-dark contrast. How will these two filters develop?

The red filter will respond to all images of red rectangles, which is roughly half of class 1 (vertical, red or blue rectangles) and will receive gradient updates increasing its selectivity for red. On images of green and blue rectangles it will likely have a negative output, which is set to 0 by the ReLU, so it will not receive gradients. On images with magenta rectangles (half of class 2), it may generate a positive output as long as it has positive coefficients in the blue channel. It will then receive gradients reducing its coefficients in both the red and the blue channel. Since red and magenta images are appear in random order, the filter will eventually end up with positive coefficients in the red channel and negative ones in the blue channel. If we imagine the NTK for only this filter, there are initially three clusters of images with similar gradients: red images, blue or green images, and magenta images. Once the filter develops negative coefficients in the blue channel, the latter two clusters collapse into one. Since the gradients for all images in one cluster point in the same direction, the filter is quickly pushed towards color selectivity.

Since every image of a rectangle contains two vertical edges, the vertical contrast filter will generate a response on every image, with the possible exception of images in which the background has a similar intensity as the rectangle. It will receive gradient updates that increase its activity on images of vertical rectangles, and decrease its activity on images of horizontal rectangles, but how these updates affect its coefficients depend on the color of the rectangle and on the intensity of the gray background. Therefore, the gradient updates for this filter do not fall into a few, clearly delineated clusters. When updates are averaged over a batch, they will interact and may partially cancel out.

Since weights are initialized randomly, most units in a network will initially have a mixed selectivity for multiple features. In the thought experiment above, it should be intuitive that such filters will quickly become color selective and lose their shape selectivity. The results of our NTK clustering analysis show that similar dynamics are at work on all four of our datasets in a larger network with multiple layers.

## Experiment 4: Spatial competition enables shape learning by suppressing clusters in the NTK

Out of all networks tested, the convolutional network with spatial competition (*spcConvNet*) was the only one that showed some selectivity for shape after training on datasets with color or texture features. To understand how this difference in bias arises, we repeated the neural tangent kernel analysis (see previous section) with this network.

We first compared the NTK of a *spcConvNet* trained on datasets with two co-occurring features (i.e., color + shape and texture + shape) to NTKs of *spcConvNet*s trained on the color-only, texture-only, and shape-only versions of the same datasets. The results are shown in Fig 9.

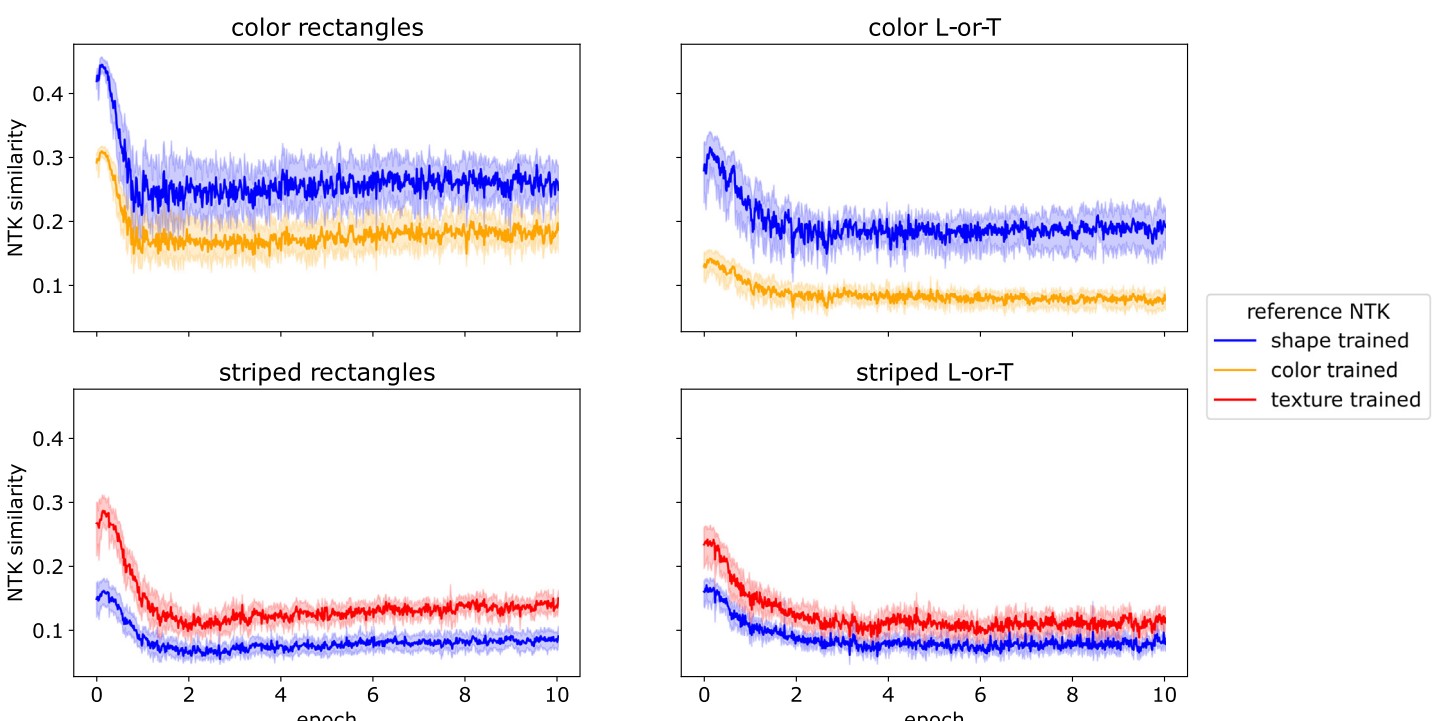

**Fig 9. NTK similarity for *spcConvNet*.** Similarity between neural tangent kernels (NTK) of trained *spcConvNet*s and of shape-selective and color-/texture-selective *spcConvNet*s throughout training. Note that the y-axis has a different scale than in Fig 7.

The NTK similarities are markedly lower than for the *ConvNet*. In the first two epochs, all similarity scores decrease and subsequently remain on a low level between 0.1 and 0.3. This lower similarity is due to the fact that the neural tangent kernels of the *spcConvNet* are sparser than those of the *ConvNet*. Essentially, the spatial competition in the *spcConvNet* architecture reduces the number of neurons that are active for a given input, which in turn means that the gradients for many parameters are small or zero and that the gradient vectors are sparser overall. This, in turn, reduces that probability that the gradient vectors for two images will have a large inner product, such that the NTK is also sparser. Thus, the network does not learn color, texture, or shape features for the whole datasets. Instead, it learns a larger mix of features, each of which applies to a subset of images—some of which related to color or texture, others to shape.

The sparsity of the NTKs also reduced the prevalence of image clusters with similar gradients. We ran the same clustering analysis on *spcConvNet* as on *ConvNet* (Experiment 3), i.e., we used the NTK as an affinity matrix for spectral clustering and measured the agreement between the resulting image clusters and true image classes using the adjusted Rand index. The results are shown in Fig 10. The agreement between clusters and image classes remained low for all three NTK types (shape, color, and texture), irrespective of the number of clusters. Comparing the maximum agreement between different NTK types with Brunner-Munzel tests revealed no significant differences on `Color Rectangles` ($\hat{p}^*(17.7) = 0.218, p = 0.830$), `Striped Rectangles` ($\hat{p}^*(17.5) = 1.035, p = 0.315$), `Color L-or-T` ($\hat{p}^*(14.4) = -2.088, p = 0.055$), or `Striped L-or-T` ($\hat{p}^*(17.8) = 0.500, p = 0.623$).

These results corroborate the idea that the presence of image clusters with similar color- and texture-selective gradient components is what biased the learning dynamics of the

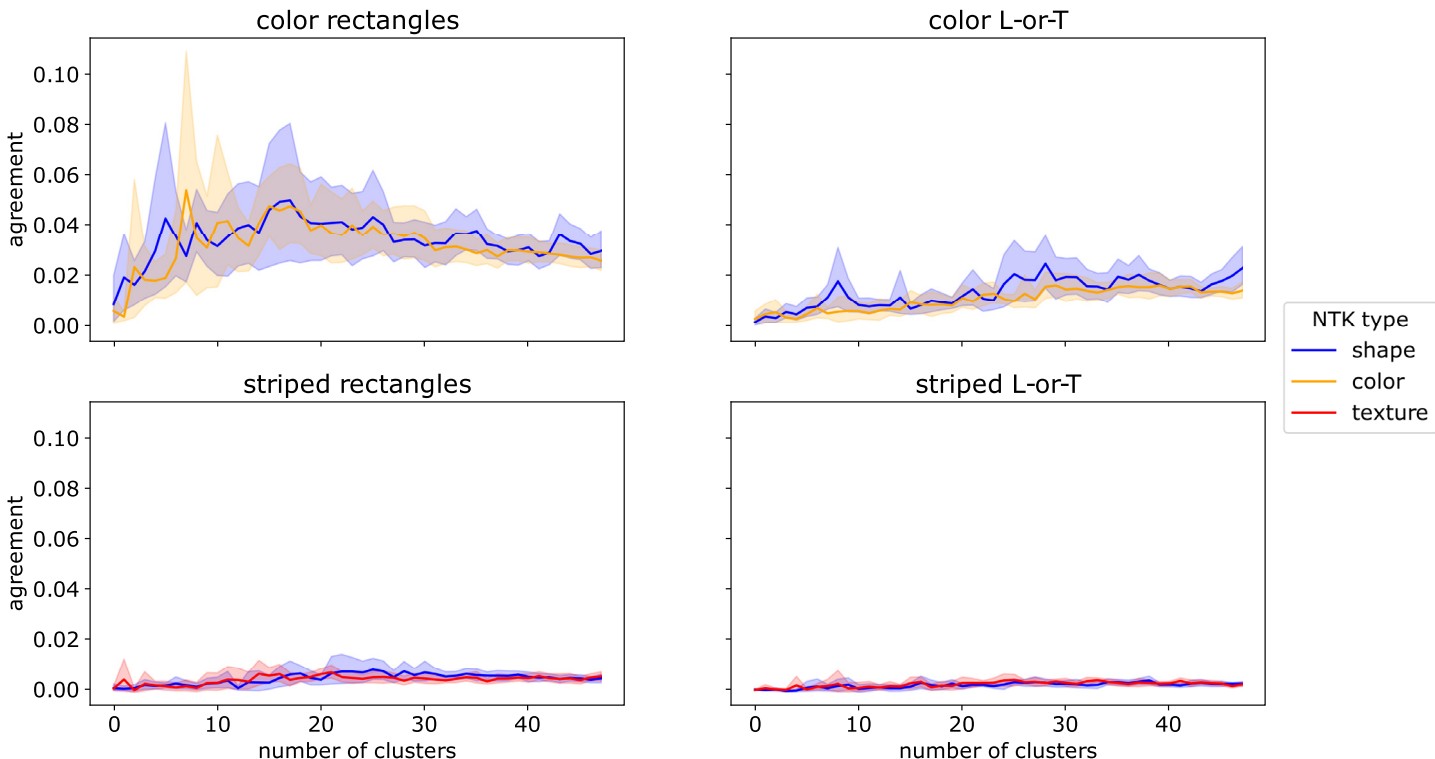

**Fig 10. NTK cluster analysis for *spcConvNet*.** Agreement (adjusted Rand index) between ground truth labels and clusters of images with high similarity in neural tangent kernels (NTKs) trained to be selective to a single feature. Note that the y-axis has a different scale than in Fig 8.

*ConvNet* towards these features. The *spcConvNet* was able to learn shape-based classification because its sparser gradient structure reduced the prevalence of these clusters.

## Discussion

The purpose of our experiments was to understand why neural networks show a preference for surface features like color and texture over shape. To this end, we designed simple artificial images in which we know (and can manipulate) the value of color, texture and shape features as well as their relationship to image class: colorful rectangles, striped rectangles, colorful L- and T-shapes, and striped L- and T-shapes. This allows us to test which features a network is biased towards and how this bias arises.

However, the simplicity of our stimuli carries a risk: there could be a qualitative difference between how networks learn simple features that are highly predictive of the target class and how networks learn more complex, ambiguous features in natural images. Even if we can explain why a network prefers color or texture over shape in our images, there is no guarantee that the same explanation applies to natural images. Nevertheless, we think that understanding why one feature and not another is learned on our simple images can be a valuable first step towards understanding the complex learning processes on natural images.

To reduce the risk that our results only reflect the idiosyncrasies of our artificial stimuli, we designed datasets with two very different types of shape features. Distinguishing horizontal and vertical rectangles requires a global shape feature (aspect ratio), whereas distinguishing Ls and Ts is similar to the relational deformations explored in [37], but can be solved with local shape features (detecting L- and T-junctions). We also repeated our experiments with a slightly more complex dataset with more different shapes (rectangles, ellipses, triangles, crosses, and parallelograms), which we refer to as the multi-shape dataset—see S1 Appendix. The results were similar to experiments 1 and 2, demonstrating that the phenomena we describe here are not limited to our specific choice of datasets. The multi-shape dataset included both local shape features (such as different angles) and global features (aspect ratio) and we tested whether there were differences in which features the networks could learn (see Fig C in S1 Appendix). There were no clear differences between the different types of shape features, neither between the rectangle and L-or-T datasets, nor on the multi-shape dataset. In contrast, previous work showed that deep networks typically succeed at learning local shape features, but struggle with global shape [17, 20, 22]. This difference may be due to the simplicity of our datasets. While the distinction between horizontal and vertical shapes is global in the sense that it concerns the overall shape, it does not require detecting the correct spatial relation of multiple object parts.

Our first set of experiments (**Experiment 1**) show that networks trained on our simple images exhibit the same biases that have been documented for natural images. We trained networks on images which could be classified by shape and a second feature, which could either be color or texture, both of which are usually characteristic of local image regions or surface patches. With the exception of *spcConvNet*, all network architectures consistently learned the second feature and disregarded shape. This mirrors the behavior of networks trained on ImageNet when they are tested with cue conflict images which have the shape of one object category but the texture of another [17, 18]. It is also in line with results from other, more complex artificial image datasets [19]. The consistency of these biases across different image types suggests that there is a common underlying mechanism. While it is conceivable that there is one explanation for why networks are biased towards texture in simple images and another, completely independent explanation for why networks are biased towards texture in complex, natural images, we believe this is unlikely.

The second set of experiments (**Experiment 2**) tested whether networks could learn to classify images by shape if there was no second feature to bias the learning. We trained the same networks as in experiment 1 on shape-only versions of our image datasets, i.e., on homogeneously grey rectangles and homogeneously grey L-shapes and T-shapes. This allowed us to test whether the networks were biased due to shortcut learning (**hypothesis 1**) or because their architecture has some defect that prevents them from learning shape (**hypothesis 2**).

Indeed, our results depended strongly on the network architecture. The multi-layer perceptron was able to learn shape-based classification above chance level, but only barely. Surprisingly, all three transformer-based architectures that we tested completely failed to learn on the shape-based datasets when trained from scratch. In contrast, both *ViT-B-16* and *Swin-T* were able to learn the shape-only classification tasks after pre-training them on ImageNet and fine-tuning the last layer (see Fig E in S1 Appendix).

These results admit two very different interpretations. On one hand, it is possible that some aspect of our stimulus design or training setup interferes with how transformers learn. To test this, we repeated our experiments with a more complex dataset (see Fig C in S1 Appendix) and with a wider range of hyperparameters (see Fig F in S1 Appendix). Again, transformers were unable to learn classification tasks when only shape information was available. This shows that the underlying problem is at least somewhat robust. Nevertheless, it is possible that the transformers fail to learn the shape-only classification tasks not because of a deficiency in their shape processing, but because our images are too artificial and too simple. Alternatively, increasing the size of our dataset and models may make transformers more shape selective [38]. In this view, we cannot conclude anything about whether transformers can or cannot process shape in natural images.

On the other hand, our results may indicate that the transformer architecture has a deficiency that prevents it from learning shape—at least the shape features present in our dataset—supporting hypothesis 2. After all, despite their simplicity our shape-only datasets did contain a feature (shape) and other networks succeeded in learning the task. In addition, the simplicity of our datasets did not prevent the transformers from learning when color or texture features were available. The transformers seem to struggle specifically with learning shape. This is at odds with previous work in which vision transformers outperformed convolutional networks when classifying silhouettes [22] and exhibit a higher shape-bias than convolutional networks [39], especially when scaled to a large size and trained on a large dataset [38].

A possible explanation is that the higher shape bias in transformers does not mean that they use shape to classify images. Instead, they may still use local, texture-like features, but in a way that is more robust to the style-transfer-based cue conflict test used in [39]. This may be explained by the insight that the self-attention operation used in transformers is actually more akin to similarity-based grouping with relaxation labeling [40]. In this interpretation, vision transformers essentially work by finding image locations with similar or compatible features and combining their representations. For example, several patches that belong to the same object may be grouped (i.e., produce mutually high attention values), such that their representations are combined. This may explain why vision transformers show a higher shape bias [39]: they are more robust against changes in texture by image style transfer, since they may still be able to find matching patches from the original object. Similarly, they may be better than convolutional networks at classifying silhouettes [22] because they can find matching features along the silhouette outline. However, this kind of grouping by similarity is not enough to characterize the shape of an object, especially if it is not dependent on the relative spatial position of the patches that are grouped. In human vision, other spatial grouping principles are equally important [41, 42]. In our simple images (e.g., of a gray rectangle on a gray background with different intensity), similarity-based grouping adds very little useful information.

On the contrary, the shape-based classification tasks require detecting specific, unique image locations (the borders of the rectangle and the L- or T-junctions), a task in which transformers underperform [40]. In this sense, it may be the simplicity of our shape-only images that makes them hard to learn for transformers: there are very few features to be grouped and the features are exactly the same in both classes, differing only minutely in their spatial arrangement.

Thus, in both explanations the simplicity of our shape-only datasets is what prevents the transformers from learning the task. The question is whether there is a qualitative difference between how transformers process natural images and our artificial stimuli (interpretation 1), or whether transformers have an underlying deficiency preventing them from learning certain shape features, but natural images contain enough other features to mask this problem. Settling this question and examining these limitations of transformers further is an interesting direction for future work.

In contrast to transformers, convolutional networks were able to learn shape-based classification, with the exception of *ResNet-50* on the shape-only L-or-T task. This means that their bias for color and texture is due to shortcut learning and not a limitation of their architecture, at least on our simple artificial images. For each of these networks, there is a set of weights that performs shape-based classification to a high accuracy—the learning algorithm simply fails to find it.

To understand how this bias arises, we examined the learning dynamics of a small *ConvNet* through the lens of the neural tangent kernel [26] (**Experiment 3**). First, we showed that throughout training, the NTK of this network was more similar to the NTK of a color- or texture-selective reference network than to the NTK of a shape-selective one. Second, we analyzed the structure of these shape-, color-, and texture-selective reference NTKs and showed that the color- and texture-selective NTKs contained clusters of images which belonged to the same class and for which the gradients (i.e., the updates to the network weights) were highly similar. In contrast, the shape-selective NTK contained less image clusters of this kind. The presence of these image clusters offers an explanation for why learning is biased towards color and texture but not shape. As the networks are trained with gradient descent, each image influences the weights by a small amount. The weight-change contributions from many images are averaged—explicitly within a batch and implicitly across batches due to the use of a small learning rate, momentum updates, etc. If one image contains both a shape feature and a color/texture feature, the gradient update for this image may increase the selectivity of the network to both features. If there are many images for which the color/texture features lead to similar gradients, the weight updates from these images will be preserved across the averaging. In contrast, the shape features for which there are no such groups of images will be averaged out. As training progresses, the network is pushed towards color-/texture-selectivity.

This explanation illustrates why training on stylized images [18], using appearance-changing image augmentation [24], and avoiding very small random crops [24] can help increase the shape bias of deep networks, but still fails to make them sensitive to global shape features [22]. Essentially, gradient descent learning favours features with occurrence statistics that align well with the image class distribution. Image stylization makes textures less predictive of image class, making it more likely that the network learns other features. Appearance-changing augmentations like color distortions and blurring have a similar effect. In contrast, using larger and more central crops means that it is more likely that training images will contain boundaries of the central object, increasing the prevalence of shape features. However, the underlying learning dynamics are still the same, so the networks will learn features with favourable occurrence statistics. We hypothesise that local shape features (for example, a specific contour on an object outline) occur more frequently and are more straightforwardly aligned with image

classes than global shape features (for example, a specific arrangement of several object parts), biasing the network to local rather than global features.

We repeated the NTK-based analysis with *spcConvNet* (**Experiment 4**), the only network architecture that learned to classify images by shape as well as color/texture in Experiment 1. We observed a lower alignment between the trained NTK and reference NTKs, which can be explained by the observation that spatial competition makes the gradients (and in turn the NTK) more sparse. This increased sparsity also reduced the prevalence of clusters of images of the same class with very similar gradients in the color- and texture-selective NTKs.

What do these results mean for deep neural networks as models of the brain in general and for shape-selective visual processing in particular? Deep learning models describe the brain not in terms of detailed, low-level interactions, but in terms of architectures, loss functions, and learning algorithms [3, 5]. The failure of deep networks to account for human-like shape processing raises the question which of these aspects need to be corrected: do we need a better architecture, a different loss function, or a new learning algorithm? Our results suggest that the choice of architecture is important, but that the more crucial limiting factor for current convolutional networks is the learning algorithm.

An example for the importance of architecture is the contrast between vision transformers, which were unable to learn shape-based classification from scratch in our experiments, and convolutional networks, which generally succeeded to learn shape-based classification in experiment 2. However, among the convolutional networks we tested the role of architecture was less clear. In the context of shape processing, one architectural component that is often discussed is lateral recurrent connectivity [20–22]. In our experiments, recurrent connectivity did not have a major effect on shape sensitivity—the recurrent *rConvNet* essentially behaved like the feedforward *ConvNet*. This is in line with other work that found no effect [22] or at best a small advantage [21] of recurrent architectures for shape processing. There may be other compelling reasons to add recurrent connections [43, 44] and models of visual function will eventually have to account for the recurrent connectivity in cortex [5]. But our results suggest that recurrence alone does not solve the underlying problem that prevents deep networks from learning shape.

We also experimented with a second biologically inspired modification to the network architecture: spatial competition. Here, our results are more promising: the *spcConvNet* was the only network architecture that showed shape selectivity after training on the dual-feature datasets in experiment 1. This mirrors results by [25], who added a custom dropout layer to their network in order to suppress information in homogeneous areas, which has a similar effect to spatial competition. This "informative dropout" reduced the reliance of networks on texture and increased shape bias. These results are promising and lend support to recent calls for more biologically inspired components in deep networks [12, 13]. Competition is an especially promising candidate for incorporation into deep networks for several reasons: its cortical equivalent—normalization—is well documented and pervasive [45] and there are theories about the computational role it could play in vision by sparsifying and whitening representations [46, 47]. In addition, it may facilitate the inclusion of other biologically inspired components, such as long-range connections, by stabilizing the system [13, 48].

However, spatial competition alone is probably not enough to enable deep networks to process shape. While *spcConvNet* was less biased towards color and texture than other architectures, it still showed a texture-bias on the `Striped Rectangles` and `Striped L-or-T` datasets. In addition, spatial competition came with a performance penalty, even on our very simple datasets. For example, *spcConvNet* had an accuracy of 96.1% on `Striped L-or-T`, whereas the *ConvNet* performed at 100%. This indicates that spatial competition reduces the

color- and texture-bias by making it harder for the network to learn color and texture features, not by making it easier to learn shape.

Instead of such changes to network architecture, enabling deep networks to recognize shapes will likely require changes to the learning algorithm. As we have argued above, our results in experiments 3 and 4 demonstrate that the color- and texture-bias arises from the dynamics of gradient descent learning, specifically from the way that gradients for color and texture features align with image classes. Breaking this alignment either requires changing the gradients (i.e., the learning rule) or changing the classes (i.e., the loss function). One promising approach for the latter is self-supervised learning. Recent work has shown that self-supervised learning objectives can reduce texture bias [24] and predict neural representations in the ventral visual stream equally well as supervised models [49, 50]. The training objective underlying these self-supervised approaches is instance discrimination [51], essentially treating each image in the training dataset as its own class. This may help reduce the alignment-effects we observed in experiments 3 and 4. However, these learning algorithms still rely on mini-batch gradient descent, so some form of alignment is likely to dominate their learning dynamics [32, 33]. Changing these alignment dynamics may require a fundamentally different learning approach: instead of changing many weights by a small amount for each example, networks could limit each weight update to a small subset of weights, thereby limiting interference between examples. In this context, work on combining deep networks with an explicit memory mechanism [52, 53] are promising. If networks can first store training examples in an associative memory and later learn to extract common patterns from them, this may reduce the averaging effect of gradient-descent learning and help them discover rarer, more complex features like shape.

Extending deep networks with a hippocampus-like associative memory model may also be important for another reason: taking advantage of the role other brain areas play in shape recognition. So far, our discussion has assumed that neural network models of visual cortex should show the same shape selectivity exhibited by human participants. This is reasonable, since many of these models aim to describe how processing in the ventral visual stream enables object recognition [8–10, 43, 49, 50, 54]. However, some recent work has questioned whether the ventral stream itself is sensitive to object shape [55, 56]. For example, [56] showed that a simple observer model of fMRI BOLD responses in human visual cortex could distinguish between image classes, but not between natural and scrambled images. Rather than being represented directly in the ventral stream, object shape may be computed by interactions between the ventral and dorsal visual streams [55, 57] or with an additional decoding step [56]. An interesting candidate would be the medial temporal lobe and hippocampus, which could receive ventral stream representations as an input and store unique features of an image [53], such as the global configuration.

If the ventral stream may be less sensitive to shape than previously assumed, does the lack of shape sensitivity in deep learning models matter? In our opinion it does. A core goal of using deep networks as models of the brain is to build an explanatory bridge that connects task constraints, neural data, and behavior [3–5]. Thus, these models ultimately need to explain what gives rise to the shape sensitivity which humans show behaviorally.

While we have only examined the bias for shape, color and texture, there are several other differences between deep networks and human vision [12], such as the susceptibility for adversarial attacks [14, 15] and robustness to image degradation [6, 16]. While our results have no direct bearing on these issues, analyzing the learning dynamics in a similar way is an interesting direction for further work. For example, it has already been shown that networks can learn to be robust to certain image degradations when explicitly trained for them [6], similar to how networks can learn shape-based classification on a shape-only dataset. While each of these

problems can be solved, for example by designing custom architectures [58, 59] or training on augmented datasets [60, 61], understanding how these differences to human vision arise from the learning dynamics of deep networks may point the way to unified solutions.

In summary, we have used simple, artificial images to investigate why deep neural networks trained to classify images develop a bias towards surface features like color and texture, instead of using shape features. While some architectures (for example vision transformers) appear to be structurally unable to learn shape features, convolutional neural networks developed this bias even though they were able to learn shape-based classification. Thus, their bias arises from the learning dynamics, not from a problem with their architecture. Furthermore, by using the neural tangent kernel as an analytical tool we identified a mechanism by which the learning dynamics of mini-batch gradient descent can bias a network towards color and texture: there are groups of images for which gradients that increase the sensitivity to these features are aligned, whereas there are no such image groups for shape features. These insights open up new opportunities to improve deep network models of visual processing in the brain.

## Materials and methods

### Datasets

We generated four datasets of simple, artificial images, as described in the Results Section. Each image shows a single object which belongs to one of two classes. The class of each object can be predicted by two features. One of these features was always the shape of the object (in dataset 1 and 2: whether a rectangle was horizontal or vertical; in dataset 3 and 4: whether the object was an L or a T). The other feature could either be color (in dataset 1 and 3) or the orientation of a striped sine grating texture (dataset 2 and 4). Since both features are predictive of image class, a network can learn either feature or both, similar to [19]. Using different combinations of shape and color / texture, we are able to test which feature each network has learned to use for classification.

We generate two version of each dataset, one with small images ($18 \times 18$ pixels) and one with larger images ($112 \times 112$ pixels). This enabled us to train and analyze a set of small networks in detail (on the small images), but also test the performance of widely used standard architectures (on the large images).

**Color rectangles.** The first dataset consisted of rectangles with different colors. Each image showed a single rectangle, which could belong to one of two classes. Rectangles in class 1 were horizontal and had red or blue color. Rectangles in class 2 were vertical and had green or magenta color. These features and the resulting class boundaries are shown schematically in Fig 11.

Specifically, the shape class depended on whether the width of the rectangle was at least one pixel larger than its height (resulting in a horizontal rectangle) or vice versa (vertical rectangle), see Fig 11A. In the small version, the longer side could be 7 to 12 pixels long. The shorter side could be 5 to 9 pixels long, but was always at least 1 pixel shorter than the long side. This yielded 24 possible combinations of side lengths per class, resulting in 48 possible rectangle shapes. To generate images, each rectangle shape was placed at each possible position in the interior of the $18 \times 18$ pixel image, excluding a 2-pixel wide border at the edge of the image. Leaving this border ensures that the outline of the rectangle can be detected and avoids ambiguity about whether the rectangle extends beyond the image.

For each rectangle shape and each possible position, we generated 5 images with different colors by randomly sampling the values of the $r$ (red), $g$ (green), and $b$ (blue) channel according to the color class (see Fig 11B). For rectangles in class 1, the green channel was drawn uniformly from the interval $g \approx [0, 1]$. The red and blue channels were either drawn from the

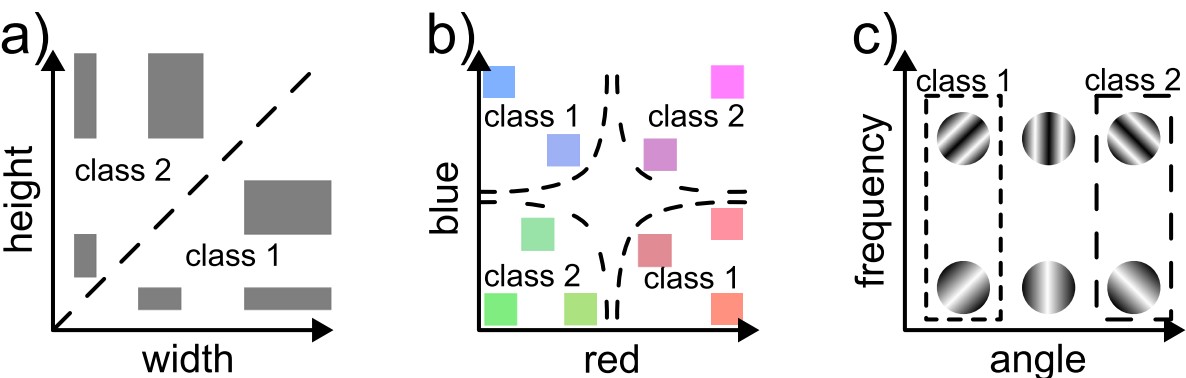

**Fig 11. Feature dimensions used in dataset design.** Three of the features used in designing datasets. Dashed lines show class boundaries in feature space. **a)** Rectangle shape classes are based on the orientation or aspect ratio. **b)** Color classes are based on the value of the red and blue color channels. **c)** Texture classes are based on the angle of a sine grating texture. The spatial frequency of the textures varied across images, but did not affect the image class. No feature dimensions are shown for the L-or-T class distinction because these classes are categorical (see text).

intervals $r \approx [0.9, 1]$ and $b \approx [0, 0.1]$ or vice versa: $r \approx [0, 0.1]$ and $b \approx [0.9, 1]$. Thus, either red had high intensity and blue low intensity or the other way around. Which channel had the high intensity was chosen randomly with probability 0.5 for either option. The red and blue channels are effectively in a logical XOR relationship. For rectangles in class 2, the green channel was again drawn uniformly from the interval $[0, 1]$. If $g < 0.5$, both red and blue were drawn from the interval $[0.9, 1]$. If $g \geq 0.5$, both red and blue were drawn from the interval $[0, 0.1]$.

The two color classes effectively form an XOR classification problem in RGB space, making the classification non-linearly separable. A simpler, linearly separable set of color classes might bias the networks towards color simply because it is trivial to learn.

Each image had a grey background ($r = g = b$) with a random intensity drawn from the interval $[0.1, 0.9]$. Note that we kept the intensity range of the background separate from the ranges for the $r$ and $b$ channels of the color classes. This simplifies the construction of color-selective networks later on. The total number of images is 10250.

For the large version of this dataset, the generation procedure was largely identical, except that the possible values for rectangle side lengths were 20, 30, 40, 50, and 60 for the longer side and 10, 19, 29, 39, and 49 for the shorter side. In addition, we did not generate images for every possible rectangle location, but only for locations on a regular grid with a 10 pixel stride. This resulted in a total of 4260 images.

**Striped rectangles.** The second dataset consisted of rectangles with a sine grating texture. The shape classes (horizontal or vertical rectangles) and the procedure for shape generation (possible side lengths, positions, etc.) were the same as in the `Color Rectangles` dataset (see Fig 11A). The only difference was that we use texture instead of color classes.

The textures were sine gratings with frequency $f$, orientation $\theta$ and phase $\phi$. Such gratings constitute coarse, directional textures [62]. For a given pixel position $(x, y)$, the intensity was $I(x, y) = \sin(2\pi \cdot f \cdot (x \cdot \cos(\theta) + y \cdot \sin(\theta)) + \phi)$. The two texture classes (see Fig 11C) differed in how $\theta$ was sampled. For class 1, $\theta$ was sampled uniformly from $\left[\frac{\pi}{4} - \frac{\pi}{6}, \frac{\pi}{4} + \frac{\pi}{6}\right]$, resulting in forward-slanted stripes. For class 2, $\theta$ was sampled uniformly from $\left[\frac{3\pi}{4} - \frac{\pi}{6}, \frac{3\pi}{4} + \frac{\pi}{6}\right]$, resulting in backwards-slanted stripes. The spatial frequency $f$ was sampled uniformly in the range of 0.05 to 0.15 cycles per pixel. The phase $\phi$ was sampled uniformly on the range $[0, 2\pi)$.

As with the color rectangles, we generate a small version of the dataset with images of size $18 \times 18$ and a large version with $112 \times 112$ pixel images.

**Color L-or-T.**   The third dataset consisted of images of L and T shapes. Each image contained one object comprised of two connected bars, one vertical and one horizontal. The end of one bar could either touch the end of the other bar, forming an L-shape (class 1), or the middle of the other bar, forming a T shape (class 2). Bars could be 7 to 13 pixels long and 3 or 4 pixels thick. We generated an image for every possible combination of bar lengths, each of the four possible orientations of the resulting L or T shape, and each position in an $18 \times 18$ pixel image (again excluding a 2-pixel buffer at the image border), resulting in a total of 11664 images. For each image, the background was set to a random grayscale value, as described for the `Color Rectangles` dataset.

The color classes were the same as for the `Color Rectangles` dataset (see Fig 11B). Objects in the first class (L-shapes) had a red or blue color. Objects in the second class (T-shapes) had a green or magenta color.

As with the rectangle datasets, we also create a large version with images of size $112 \times 112$. Here, the height and width of each shape could be 30, 40, or 50 pixels and the bars making up the shapes could be 5, 10, or 15 pixels wide. Again, we generated images for locations with a relative offset of 10 pixels to each other. This resulted in a total of 10584 images.

**Striped L-or-T.**   The fourth dataset used the same shape classes as the `Color L-or-T` dataset and the same texture classes as the `Striped Rectangles` dataset (see Fig 11C). Thus, each image either contained an L-shaped object with a forward-slanted sine grating texture, or a T-shaped object with a backwards-slanted sine grating texture. Again, we generate a small ($18 \times 18$ pixels) and large ($112 \times 112$ pixels) version of the dataset.

**Control and conflict datasets.**   In order to test whether networks learn to use shape, color, or texture information to classify images, we created several variants of our datasets. In a **shape-only** variant, we eliminate the color or texture feature and instead render the rectangles, Ls, or Ts in homogeneous gray with $r = g = b = 0.5$. All other features of the dataset (possible heights, positions, etc.) stay the same. In a **color-only** or **texture-only** dataset, we eliminate the shape feature by replacing the rectangles, Ls or Ts with squares that contain the appropriate color or texture.

In a **conflict** variant of a dataset, we reverse the assignment of colors or textures to classes. For example, in the original colored rectangle dataset, horizontal rectangles are red or blue and vertical rectangles are green or magenta. In the conflict variant, this relationship is inverted, so horizontal rectangles have color green or magenta color and vertical rectangles have color red or blue color.

By testing a network on a shape-only, color-only, or texture-only variant, we can test whether it can learn to use the respective feature. If it failed to learn the feature during training, accuracy should drop to chance. By testing a network on a conflict variant, we can test whether a network is biased to shape or color/texture in its decisions.

## Network architectures

A lack of shape sensitivity has been documented for a wide variety of networks [17, 18, 21, 22]. To understand which role the network architecture plays in shape processing, we test a range of networks on our datasets, including a multi-layer perceptron, several convolutional neural networks, and vision transformers. We start with a set of **basic networks**, which we deliberately keep small to be able to analyze them in more detail. We also test a set of **standard networks**, which are widely used.

**Basic networks.** We evaluated a multi-layer perceptron, a convolutional neural network, and a vision transformer, as well as convolutional network variants with recurrent connectivity and spatial competition.

The multi-layer perceptron (**MLP**) had 3 fully-connected hidden layers, each with 1024 units. Each fully-connected layer was followed by layer normalization [63] and a GELU non-linearity. The last layer of the network was a linear fully-connected layer with 2 output units.

The convolutional network (**ConvNet**) contained 3 convolutional layers, each with kernel size $5 \times 5$. The number of output channels increased from 16 in the first layer to 32 in the second and 64 in the third. Each convolution layer was followed by group normalization [64], where the number of groups was equal to the number of channels, and a GELU non-linearity. After the last convolution layer, the output was flattened into a vector and fed to a linear, fully-connected layer with 2 output units.

Notably, the *ConvNet* did not contain any max-pooling layers. We chose this design because strided pooling as used in most convolutional networks may destroy some shape information that is required for the tasks we designed. For example, shape-based classification of rectangles may require detecting a 1-pixel difference between the height and width of a rectangle. Since strided pooling makes the network output more invariant to changes in input size, it may destroy a network's ability to perform this classification.

The vision transformer (**ViT**) split the image into 36 non-overlapping patches of $3 \times 3$ pixels. Each patch was linearly projected to a 128-dimensional embedding. The resulting tokens and an additional classification token were passed through 12 encoder layers with a hidden dimension of 1024 units and 8 heads. The classification token was mapped to two output units with a linear fully-connected layer.

Recurrent connections play a crucial role in shape processing in visual cortex [42, 65–68] and have been proposed as an important component in making deep networks more biologically plausible [44, 54, 69–72]. Therefore, we also included a convolutional network with recurrent connections (**rConvNet**). This network had the same architecture as the feedforward *ConvNet* with the addition of recurrent connections in each convolution block. Specifically, each block contained two convolutional kernels $K_{ff}$ and $K_{lat}$, which were applied to the current feedforward input $x[t]$ and the block output $y[t-1]$ from the previous time step:

$$y[t] = \text{GELU}(\text{GN}(K_{fw} * x + K_{lat} * y[t-1] + b)) \tag{1}$$

where $b$ is a bias term, $*$ denotes convolution, and GN denotes group normalization. On the first time step, $y[0]$ was 0. The network was run for 10 time steps and trained with back-propagation through time.

Another common feature in models of biological shape processing is spatial competition [13, 65, 66]. For this reason, we also included a convolutional network with a simple spatial competition mechanism (**spcConvNet**) in our analysis. Like *ConvNet*, *spcConvNet* consisted of 3 blocks with 16, 32, and 64 feature maps, respectively, and a final fully-connected layer. However, each convolution block consisted of a depth-wise convolution (1 kernel per input channel, no interactions between channels), followed by group normalization with as many groups as there are input channels. Spatial competition is implemented with a local softmax operation:

$$s(x_{i,j,c}) = \frac{\exp(x_{i,j,c}/\tau)}{G_{i,j} * \exp(x_{i,j,c}/\tau)} \tag{2}$$

where $i$ and $j$ index spatial dimensions, $c$ indexes channels, $G_{i,j}$ is a Gaussian kernel, and $*$

denotes convolution. This operation puts nearby locations within a feature map into competition, implementing a divisive on-center off-surround interaction. The strength of the competition is controlled by the temperature parameter $\tau = 0.2$. The spatial competition is followed by a $1 \times 1$ convolution that maps input channels onto output channels per pixel. The number of output channels in each block is the same as in the feedforward *ConvNet*—16 in the first block, 32 in the second, and 64 in the third. Local response normalization [73] is applied to the output.

**Standard networks.** To make sure that our results are not limited due to the smallness of the networks we trained on the $18 \times 18$ datasets, we also trained several widely used network architectures on the dataset versions with larger images of $112 \times 112$ pixels. Specifically, we trained *VGG-19* [27], *ResNet-50* [28], the vision transformer *ViT-B-16* [29], and a sliding-window vision transformer *Swin-T* [30]. For each model, we used the standard implementation in the torchvision library (version 0.16.0). For *VGG-19*, we used *vgg19_bn*, the modernized version with batch normalization implemented in torchvision.

## Training

We trained each network from scratch on each dataset. Each network was trained for 30 epochs with a stochastic gradient descent optimizer using a momentum term with $\gamma = 0.9$ and Nesterov's accelerated gradient. The initial learning rate was set to 0.01 and reduced according to an exponential decay learning rate schedule with factor 0.99 at the end of each epoch. In addition, we used a weight decay of 1e-4 for regularization. We used the cross-entropy loss function to train the networks for 2-class classification.

All networks and training loops were implemented in PyTorch [74] (version 2.1.0) and pytorch-lightning [75] (version 2.1.0).

## Testing network selectivity

We evaluated network performance on our artificial image datasets to answer two types of questions: (1) can a given network learn to use a specific feature to classify images (**learnability**) and (2) if a network is faced with two equally predictive features during learning, which one does it rely on (**bias**).

To answer the question of **learnability**, we train the network on a dataset in which only a single feature is available and test it on a test set with the same feature. For example, we train the network to distinguish the letters L and T, rendering each letter in homogeneous grey (no color and no stripe textures). If the network learns this task, it must be able to use the shape feature.

To test whether a network has learned a task, we compare the test accuracy of $n$ independent training runs against chance. Since all our tasks are 2-way classification tasks, chance level is an accuracy of 0.5. We perform a sign test to check if performance significantly differs from chance. The test statistic is:

$$M = \frac{k^+ - k^-}{2} \tag{3}$$

where $k^+$ is the number of training runs with an accuracy above 0.5 and $k^-$ is the number of runs with an accuracy below 0.5. Under the null hypothesis that the network performs at

chance level, $k^+$ and $k^-$ follow a binomial distribution, resulting in a two-sided p-value of:

$$p = P(k \leq k^{\min}) = \sum_{i=1}^{k^{\min}} \binom{n}{k} 0.5^k (1 - 0.5)^{n-k} \tag{4}$$

with $k^{\min} = \min(k^+, k^-)$ and $n = k^+ + k^-$.

To answer the question of **bias**, we train the network on a dataset in which two features are available and predict the image class equally well. Thus, the network could learn to use either feature, or both. For example, a network trained to distinguish L-shapes with forward-slanted stripes from T-shapes with backward-slanted stripes could use shape (L or T) or texture (stripe orientation) to classify the images. We then test the network on one dataset in which only the first feature is available (e.g., homogeneous gray Ls and Ts), one on which only the second feature is available (e.g., squares with forward- or backward-slanted stripes), and one conflict dataset where both features are available but the values of one feature are inverted between classes (e.g., L-shapes with backward-slanted stripes and T-shapes with forward-slanted stripes).

If the network learned to use both features, we would expect above-chance accuracy on both single-feature test sets and chance performance on the conflict set. In contrast, if the network learned to use one feature but not the other, we would expect above-chance performance on one single-feature test set and chance performance on the other. Performance on the conflict set should be either above or below chance, depending on which feature was learned. Again, we use a sign test to determine if performance is significantly different from chance.

Since we are interested in the overall pattern of performance of each network across several test sets, positive and negative test results are equally informative. For this reason, we do not correct for multiple comparisons.

All statistical tests were carried out in Python using the statsmodels library (version 0.14.1) [76].

## Learning dynamics and empirical neural tangent kernel

We examine the neural tangent kernels (NTKs) of networks during training to characterize how and why they become selective to specific features. While neural tangent kernels were developed to study neural networks in the infinite width limit [26], they are also useful to characterize the learning dynamics of networks with finite width [31–33]. In this setting, the NTK evolves while the network is trained and the changes in the NTK reflect feature learning in the network. Typically, there is an early phase in which the NTK changes rapidly [31] and aligns to the target function [32, 33]. Afterwards, the evolution of the NTK slows down [31] and learning can approximately be described as kernel regression with the learned, data-dependent NTK [31, 32], similar to the infinite-width case.

**NTK similarity.**   We want to understand which features a given NTK (and therefore the network) is sensitive to. To this end, we compare the NTK to reference kernels with known feature selectivity. We generate these reference NTKs by training a network on a dataset with only a single feature. For example, to get a shape-selective reference NTK for a *ConvNet* trained on `Color Rectangles`, we train a *ConvNet* on grayscale rectangles and compute the NTK after 30 training epochs.

For a given dataset $D = \{x_1, \ldots x_n\}$ of images $x_i \in \mathbb{R}^d$ and network $f(x; \theta) : \mathbb{R}^d \to \mathbb{R}^C$ with trainable parameters $\theta$ and $C$ output classes, we compute the traced NTK:

$$\Theta(x_i, x_j; \theta) = \frac{1}{C} \sum_{c=1}^{C} \nabla_\theta [f(x_i; \theta)]_c \cdot \nabla_\theta [f(x_j; \theta)]_c \tag{5}$$

where $\nabla_\theta[f(x;\theta)]_c$ is the gradient of the $c$-th output unit of network $f$ with respect to weights $\theta$ on input image $x$.

We compare kernels using their cosine similarity across all images in the dataset $D$:

$$S_{\cos}(\Theta_1, \Theta_2; D) = \frac{\langle \Theta_1, \Theta_2 \rangle_F}{\|\Theta_1\|_F \cdot \|\Theta_2\|_F}, \tag{6}$$

where $\langle\rangle_F$ is the Frobenius inner product and $\|\|_F$ is the Frobenius norm. This similarity measure is similar to the kernel alignment metric used in [33], except we use it to compare two NTKs rather than to compare an NTK to the class labels. It is also the inverse of the kernel distance used in [31]. While training a network, we repeatedly compute the similarity between its NTK and the reference NTKs of the features of interest to trace how selectivity for the different features develops. For comparison, we also measure the similarity between the trained kernel and the class labels (i.e., a kernel with value 1 if both images belong to the same class and value 0 if they belong to different classes).

**NTK structure.** Ultimately, our goal was to understand why networks learned one feature (e.g., color or texture) more readily than another (shape). Since this feature preference was reflected in the similarity between the trained NTK and feature-specific reference NTKs, we reasoned that the structure of these reference NTKs may explain why a feature is easier or harder to learn.

A common feature of the evolution of the data-dependent NTK is *kernel alignment*, the tendency of the NTK to align with the target function [32, 33], i.e., to assign a high similarity to inputs that belong to the same class and a low similarity to inputs that belong to different classes. Therefore, we are interested in how well the similarity structure of the reference NTK for a specific feature matches the true distribution of image classes. That is, we ask whether two images $x_i$ and $x_j$ that are similar according to the NTK (i.e., if the inner product of their gradients is large) usually belong to the same class. Conversely, if two image belong to the same class, are they similar according to the NTK?

To answer this question, we cluster the dataset according to the NTK. We transform the NTK coefficients to a non-negative affinity matrix $A$ using the traced NTK in Eq 5:

$$A_{i,j} = \exp\left(\frac{\Theta(x_i, x_j; \theta)}{\sigma_\Theta}\right) \tag{7}$$

where $\sigma_\Theta$ is the standard deviation over the coefficient values of $\Theta$. We used this affinity matrix to perform spectral clustering [77] using the scikit-learn library [78]. Since the quality of the clustering may change with the number of clusters, we vary the number of clusters from 2 to 50. For each resulting set of clusters, we compute the adjusted Rand index [34] between the cluster assignments and true image classes. The Rand index rates the agreement between two sets of labels and corresponds to the probability that, two randomly chosen elements (in our case: images), both sets of labels agree whether these elements belong to the same class or to two different classes. The adjusted version corrects this probability for the amount of agreement expected by chance.

To test if the NTK for one feature (e.g., color) has a significantly higher agreement with the class labels than the NTK for another feature, we train 10 independent versions of each feature-specific NTK with different random initial parameters. For each resulting NTK, we run the clustering and choose the number of clusters that results in the highest adjusted Rand index. Thus, we get 10 agreement scores per feature. We compare these two distributions using a Brunner-Munzel test [35, 36].

## Supporting information

**S1 Appendix. Additional experiments.** To demonstrate the robustness of our findings, we show results from additional experiments with a more complex dataset and with pretrained networks, as well as additional training runs with a vision transformer with a wider range of hyperparameters.
(PDF)

## Acknowledgments

The authors acknowledge support by the state of Baden-Württemberg through bwHPC.

## Author Contributions

**Conceptualization:** Christian Jarvers, Heiko Neumann.

**Data curation:** Christian Jarvers.

**Formal analysis:** Christian Jarvers.

**Funding acquisition:** Heiko Neumann.

**Investigation:** Christian Jarvers.

**Methodology:** Christian Jarvers, Heiko Neumann.

**Project administration:** Heiko Neumann.

**Software:** Christian Jarvers.

**Supervision:** Heiko Neumann.

**Visualization:** Christian Jarvers.

**Writing – original draft:** Christian Jarvers.

**Writing – review & editing:** Christian Jarvers, Heiko Neumann.

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
