## [Decision Letter · Decision Letter 0]

21 Jun 2024

Dear Mr Jarvers,

Thank you very much for submitting your manuscript "Teaching deep networks to see shape: Lessons from a simplified visual world." for consideration at PLOS Computational Biology.

As with all papers reviewed by the journal, your manuscript was reviewed by members of the editorial board and by several independent reviewers. In light of the reviews (below this email), we would like to invite the resubmission of a significantly-revised version that takes into account the reviewers' comments.

We cannot make any decision about publication until we have seen the revised manuscript and your response to the reviewers' comments. Your revised manuscript is also likely to be sent to reviewers for further evaluation.

Sincerely,

Ming Bo Cai

Academic Editor

PLOS Computational Biology

Andrea E. Martin

Section Editor

PLOS Computational Biology

Reviewer's Responses to Questions

**Comments to the Authors:**

Reviewer #1: I really enjoyed reading this paper and I think it makes important contributions to understanding differences between deep neural networks and human vision. I do have a few concerns that I would like the authors to address before the article is accepted.

First, I think the two shape comparisons that the authors tasked the network with making have interesting differences from each other. The rectangle task involves sensitivity to a more global shape feature (aspect ratio), but the shapes are different from each other only because of their orientation relative to the image frame. The L’s and T’s differ in terms of their local features (for example, number of corners). A human observer spontaneously asked whether a horizontal and vertical rectangle have the same shape might say yes, but they would almost certainly not say yes for the L and T. I don’t think the authors need to change the stimuli they used in any way, but it would be interesting to see the authors discuss how stimulus differences affected performance in their various experiments.

A more minor suggestion would be for the authors to add a paragraph to their Experiment 1 Results section explaining spatial competition architectures a little more. This may be my ignorance, but I was not very familiar with them and was curious which of their properties may be conferring performance advantages in Exp. 1 and beyond. The authors do address this in the Discussion and give more detail in the Methods section, but knowing a little more about spatial competition architectures early on would be helpful for understanding the authors’ findings.

For Figure 7 and 9, I was a little confused about what the “labels” line was plotting. Could the authors clarify?

Another small concern I had was in the way the authors discussed the two possibly explanations for color/texture bias in lines 201-206. It seems that there is ample evidence that even ImageNet-trained deep networks learn some shape features and can classify by them. Otherwise, how else would DNNs perform better than chance on silhouette images? In my view, this does not preclude the possibility that there are kinds of shape features important to human perception that are DNNs to which DNNs are insensitive, such as more global and/or configural shape features. It would be helpful if the authors clarified their view on this.

In a similar vein, I was a bit puzzled by the authors’ finding that ViT was completely unable to learn to classify by shape. As the authors discuss in lines 405-406, this is inconsistent with previous findings that ViT significantly outperforms both VGG and ResNet in silhouette classification. Unless I misunderstood, I don’t think the explanation they give in lines 406-419. What similarity-based grouping principles would allow for accurate silhouette recognition of animals in Baker & Elder (2022) but not for L’s vs. T’s?

Minor:

Line 12: “…how they have to be connected, to high-level…” should be “…how they have to be connected, and high-level…”

Line 66: “…-why networks prefer rely on texture” should be “why networks rely on texture”

Line 105: “…to lean shape-based classification” should be “to learn shape-based classification”

Line 324: “…suggest that color and shape belong to the former kind of feature…” should be “…suggest that color and texture belong to the former kind of feature…”

Reviewer #2: I enjoyed reading this manuscript. It presents a good set of studies that try to systematically understand the origin of texture/color-bias in Deep Neural Networks (DNNs). While other studies have investigated this texture / shape bias before, this study had two innovations that take the research forward: (1) it used extremely simple design to disentangle whether the network can learn simple shapes like squares and T-shaped figures presented in conjunction with other predictive features, and (2) it used a Neural Tangent Kernel to gain insight into why some features are learned better than others. The field needs more studies of this type that used controlled experiments to tease apart properties of DNNs.

There are some shortcomings to the manuscript in it's current form. I believe that the authors can address these shortcomings in a revision. Some of my suggestions are pretty straightforward to address, while others may need a bit more time.

Major comments:

1. One of my key concerns is the clarity of the manuscript, especially Experiments 3 and 4. I believe most of the readers of this manuscript will (like me) not be familiar with Neural Tangent Kernels. While the authors have done a good job, especially in the Methods, to educate the reader, there are several aspects that are still quite obscure. Here are a few examples of statements / phrases that are difficult to make sense of without a deeper understanding of the NTK. I have included the difficulty in parsing these phrases within parentheses:

l.259: "the evolution of the network function f(x;θ) follows the kernel gradient of the NTK" (but the reader may not know what a kernel gradient is)

l.260: "This is especially useful in the limit of infinitely wide neural networks" (But the authors are not considering infinitely wide neural networks. So, it's unclear to me why the authors mention this.)

l.260: "at each point during the training the direction of the kernel gradient reflects what the network is learning [25], whereas the structure of the kernel reflects the features that the network has learned" (Why is there a dichotomy between the direction and structure of the gradient?)

l.287: "Why is the NTK more similar to the color- or texture-trained network?" (This is a shortening, right? My understanding of what you mean is: Why is the NTK of the network trained on the full dataset more similar to the NTK of the color- or texture-trained network, than the shape-trained network?)

There are many other examples in the manuscript. I would recommend a substantial rewrite of Experiments 3 & 4 to clarify the language and only use jargon that is strictly necessary.

2. Related to point (1) above is clarifying some methods that the authors have used for the NTK analysis. This analysis is presented in a way that seems quite convoluted and I think many readers will find this impenetrable as there are a number of steps underlying this analysis. I would recommend that the authors present the overall structure of their argument somewhere early on in their manuscript, so that the reader knows where this is going. Here's my understanding of what this analysis involves: The analysis consists of four steps. First, the authors train three NTKs as reference NTKs. These are color-only NTK, texture-only NTK and shape-only NTK. Second, they compute a similarity (across all images in the datasets) between each of these NTKs and the NTK that learns on color+texture+shape. Frequently, they find a greater similarity with the color-only and texture-only NTKs. In Step 3, they cluster the images based on each NTK, varying the clusters from 2-50. In Step 4, they work out how well these clusters are "aligned" to the classes. The key result is that images clustered based on the color and texture NTKs align with class categories much better than those based on shape NTK. What does this show? According to the authors this shows that DNNs learn differently based on images that present the same shape but in different locations. In contrast, when the same color is presented in different locations, DNNs learn (adjust their weights) in a much more similar manner. In a mini-batch setting, this means that different images with same color lead to similar weight adjustment, while different images with same shape lead to different weight adjustments, which may cancel each other out. If my understanding is correct, this is a lovely and novel insight.

3. The other major shortcoming of the manuscript is the limited manner in which the authors have tested their hypotheses. There are several extensions to their study that may be useful and provide greater insight:

(i) One of the most surprising finding of the manuscript is that many standard network simply cannot learn some basic shapes. Many researchers will be unwilling to concede that the standard networks simply cannot learn shape. They may suspect, for example, this is due to the way the networks were tested - e.g. the size of datasets, the number of categories (2 categories might not be enough), hyper-parameters used (30 epochs may not be enough, the learning rule may not be appropriate, etc.) Therefore, it will be good to establish how robust these results are. Do they hold for other (larger) datasets? Do they hold for settings when dataset contains more than 2 categories? Do they hold for other settings of hyper-parameters (learning rate, number of epochs, regularization methods, learning rules)? To satisfy this sceptical reader, it will be good to carry out these tests.

(ii) My own suspicion is that the networks are not learning these shapes as the shapes only differ in global properties (the relative width & height of sides, or the relationship between parts). This is a hard task for the network as they are far more likely to pick up local features (as shown by the results of Baker et al., Geirhos et al. and some of my own work - e.g. Malhotra et al.). If, however, there were local (shape) features that could distinguish the two categories, the standard networks should be able to learn them. For example, shape in one category had curved vertices, while the shapes in another category had sharp (angular) vertices (square vs circle vs triangle). If this is the case, then it would be incorrect to say that the networks cannot learn shape per se, but that they cannot learn global shape features but are capable of learning local shape features. It will be good to test this.

(iii) The networks used by the authors have been trained from scratch. While this is useful, many studies have shown that pre-training the networks can lead the networks to acquire some biases. The authors can test this by downloading and using networks that have been pre-trained on various datasets, e.g. ImageNet and EcoSet.

(iv) The images in the dataset do not vary in scale. Are the network able to learn shape when scale is varied? This may seem counter-intuitive as one is increasing the size of the dataset and giving the network a harder problem to solve. But by making the problem harder, one may prevent the network getting stuck in local optima.

4. Further experiment (optional): It is interesting to note that in Fig 7 the NTK similarity evolves for a short period and then reaches an equilibrium (at least in three out of four panels). This is likely because the network "discovers" color as a predictive feature and weight updates become smaller once color can predict nearly 100% of image classes. It is possible that shape is harder to discover than color and had the network not been able to discover color, or if color was less predictive than shape, then the network may have kept on learning shape features. The authors can test this by designing a slightly different experiment where color is less predictive than shape. For example, if color predicted output category for 50% of images but shape predicted output category for 100% of images, then it is possible that the network kept learning shape features and eventually the similarity with shape NTK increases.

Other comments:

1. Can the authors discuss how their results (showing that ViT cannot learn shape) square up with recent results that some vision transformers show the highest level of shape bias (Dehghani,..., Houlsby et al, 2023, ICML)?

2. How do these authors explain the results of Hermann, Chen & Kornblith (2020) who showed that networks can show a shape-based learning and that texture-bias is due to pre-training?

3. Please make the 2x2 grid structure in Fig 1 & 2 (conditions in rows and columns) consistent with from the grid structure in Fig 7, 8, 9 & 10. (Fig 1 & 2 show color in row 1 and texture in row 2, while shape task 1 in column 1 and task 2 in column 2. Fig 7, 8, 9 & 10 use a different scheme).

4. It is not clear to me why Figs 7 & 9 contain comparisons with the 'labels' condition and the authors do not discuss the significance of this trace in any detail. If this is not important, perhaps it is better to just remove this trace.

5. Experiment 4: I am unable to see how the results in Fig 9 are consistent with the results in Fig 4. For example In Fig 4 (top left), the performance of the network on 'Color only' condition is near perfect. However, in Fig 9, the NTK similarity with color trained network is lowest. In contrast, in the 'Conflict' condition, the network seems to rely more on shape than color - this seems consistent with Fig 9 (top left). However, if one looks at the 'striped rectangles' condition, the network again performs above chance for shape-only condition. But in Fig 9, the NTK similarity with shape-trained network is lowest. How do the authors explain these results?

6. l. 78: neither -> none

7. l. 323-324: "Our results suggest that color and *shape* belong to the former kind of feature, whereas shape belongs to the latter." -> The first shape should be "texture".

Reviewer #3: The paper "Teaching deep networks to see shape: Lessons from a simplified visual world" addresses a core challenge in computer vision: why deep networks do not perceive and interpret shapes as humans. They try to understand the root causes behind the problem so that neural networks can be made more human-like. The authors have taken an innovative approach by simplifying the visual world to focus on fundamental shape recognition in a controlled set of experiments with popular neural network models, which provides valuable insights and helps in identification of key factors that influence shape recognition performance in neural networks. These learnings are important because they can be used for development of neural networks that are more human-like in behavior. However there are a few things in the paper that can be better discussed and improved upon:

— The study focuses on basic geometric shapes, which may not adequately represent the diverse range of shapes encountered in natural environments. The importance of training models on a wide variety of shapes to improve their generalization capabilities is well known. While the authors address that their experiments are simple in the discussion section, it might be compelling to see the dataset include diverse and complex shapes that enhance the robustness of the findings. While I appreciate the use of controlled stimuli for proof of concept, it would add value to confirm that the findings generalize to use of, say, circles or triangles or their combinations and are not limited to rectangle like structures.

— Lines 185-186 : It might be useful to identify some examples of the edge cases that lead to small improvement in shape selectivity. Such examples could even be used to support the findings.

— Lines 316-323: It might be useful to explain it through an intuitive figure or mathematical example, since it helps drive the take-home message clearly.

— While this paper does a good job in using controlled experiments to identify some root causes of why neural networks do not exhibit shape selectivity like humans, I wonder if the paper’s findings can shed light on other observable differences/bias between humans and neural networks for related tasks, or are the findings strictly shape selectivity specific?

— While there is some discussion about how the findings can be related to human visual system, it can be improved further. Explaining and expanding on how “work on combining deep networks with an explicit memory mechanism [49, 50]” in line 515 relates to the architectural and learning algorithm based findings of the paper can help. I assumed the goal of the paper was to bridge the gap between human behavior and that exhibited by neural networks, however, while the paper does a good job of providing some solid reasons of the gap between the two, I could not necessarily connect the findings to a testable solution.

**Have the authors made all data and (if applicable) computational code underlying the findings in their manuscript fully available?**

Reviewer #1: Yes

Reviewer #2: Yes

Reviewer #3: Yes

PLOS authors have the option to publish the peer review history of their article (what does this mean?). If published, this will include your full peer review and any attached files.

Reviewer #1: No

Reviewer #2: **Yes: **Gaurav Malhotra

Reviewer #3: No
---

## [Decision Letter · Decision Letter 1]

1 Oct 2024

Dear Mr Jarvers,

Thank you very much for submitting your manuscript "Teaching deep networks to see shape: Lessons from a simplified visual world." for consideration at PLOS Computational Biology. As with all papers reviewed by the journal, your manuscript was reviewed by members of the editorial board and by several independent reviewers. The reviewers appreciated the attention to an important topic. Based on the reviews, we are likely to accept this manuscript for publication, providing that you modify the manuscript according to the review recommendations.

The reviewers were basically happy to accept the paper as is. But there were some minor suggestions. Because typically only typeset update is allowed after acceptance, I made the decision as "minor revision" to leave it up to the authors to decide if you want to take these suggestions and make any update. The returned manuscript will be read by the editors only and won't be sent back to reviewers.

Sincerely,

Ming Bo Cai

Academic Editor

PLOS Computational Biology

Andrea E. Martin

Section Editor

PLOS Computational Biology

The reviewers were basically happy to accept the paper as is. But there were some minor suggestions. Because typically only typeset update is allowed after acceptance, I made the decision as "minor revision" to leave it up to the authors to decide if you want to take these suggestions and make any update. The returned manuscript will be read by the editors only and won't be sent back to reviewers.

Reviewer's Responses to Questions

**Comments to the Authors:**

Reviewer #1: The authors have addressed by concerns. I think this is a really nice study.

Reviewer #2: The authors have done an excellent job in revising the manuscript. If they get a chance to revise the manuscript further before publication, my one recommendation will be to more clearly state from the beginning that the networks have to be untrained for them to fail on the shape-only task and pre-trained networks are able to learn this task (e.g. in the Abstract, the authors could say that "we show that some *untrained* networks are unable to learn about shape"; similarly in the Introduction). This is useful information for future research. Other than this, I think the manuscript looks great - it is well written and informative for researchers in the area.

Reviewer #3: The authors did extend their experiment to larger range of shapes. Their results do not suggest big difference between what was previously observed in Experiments 1 and 2.

I support the decision to include the results in the supplementary and mentioning it in the discussion section. I think including these additional experiments and comparing the results with results about local-global shape learning capabilities of neural networks from recent studies is interesting but very short. I would have appreciated more discussion on how this connects to natural images, rather than a line mentioned in supplementary.

I understand the difficulty expressed by the authors about identifying “specific edge cases”. I appreciate the change of writing and better explaining this part. However, I was expecting that if the same experiment is repeated multiple times and certain images are correctly classified above chance across all the repetitions, then those could serve as potential edge case examples and might have given insights about working of the model, specifically what features lead to more shape selectivity. However, it is quite possible that such images do not exist and the effect is only an aggregate effect. In that case finding singular examples might be hard, I agree.

The example explaining analysis of the learning dynamics is very helpful for any reader who would want to develop an intuition of the learning dynamics. It also benefits readers who may not want to dive deep into the mathematical intricacies of the model.

Adding two more observed differences between humans and neural networks is great. However, I hoped there could be examples added with some more experimental simulations involved. Additionally, there are multiple papers in the literature that have explored the mentioned differences of which only one was cited for each. Also, it might not be fair to mention observations from natural image dataset experiments without necessarily testing them for simple shape dataset used in this work. The fact that authors do mention that their results do not have direct bearing to the observations can be considered cautionary, but I would probably remove this addition from the text. It feels half baked, not well referenced (there are already multiple papers on proposing principled solutions) and not relevant for the very simple dataset used here.

Thanks for improving the explanation of how memory mechanisms can enable more shape selectivity.

**Have the authors made all data and (if applicable) computational code underlying the findings in their manuscript fully available?**

Reviewer #1: Yes

Reviewer #2: Yes

Reviewer #3: Yes

PLOS authors have the option to publish the peer review history of their article (what does this mean?). If published, this will include your full peer review and any attached files.

Reviewer #1: No

Reviewer #2: **Yes: **Gaurav Malhotra

Reviewer #3: No

Figure Files:

Data Requirements:

Reproducibility:

References:

---

## [Editor Report · Decision Letter 2]

29 Oct 2024

Dear Mr Jarvers,

We are pleased to inform you that your manuscript 'Teaching deep networks to see shape: Lessons from a simplified visual world.' has been provisionally accepted for publication in PLOS Computational Biology.

Best regards,

Ming Bo Cai

Academic Editor

PLOS Computational Biology

Andrea E. Martin

Section Editor

PLOS Computational Biology

Feilim Mac Gabhann

Editor-in-Chief

PLOS Computational Biology

Jason Papin

Editor-in-Chief

PLOS Computational Biology

---

## [Editor Report · Acceptance letter]

5 Nov 2024

PCOMPBIOL-D-24-00490R2 

Teaching deep networks to see shape

Dear Dr Jarvers,

I am pleased to inform you that your manuscript has been formally accepted for publication in PLOS Computational Biology. Your manuscript is now with our production department and you will be notified of the publication date in due course.

With kind regards,

Lilla Horvath
